# Delayed Gradient Averaging: Tolerate the Communication Latency in Federated Learning

**Ligeng Zhu**[1]     **Hongzhou Lin**[2]     **Yao Lu**[3]     **Yujun Lin**[1]     **Song Han**[1]

[1]MIT     [2]Amazon     [3]Google

https://dga.mit.edu

## Abstract

Federated Learning is an emerging direction in distributed machine learning that enables jointly training a model without sharing the data. Since the data is distributed across many edge devices through wireless / long-distance connections, federated learning suffers from inevitable high communication latency. However, the latency issues are undermined in the current literature [15] and existing approaches such as FedAvg [27] become less efficient when the latency increases. To overcome the problem, we propose **D**elayed **G**radient **A**veraging (DGA), which delays the averaging step to improve efficiency and allows local computation in parallel to communication. We *theoretically* prove that DGA attains a similar convergence rate as FedAvg, and *empirically* show that our algorithm can tolerate high network latency without compromising accuracy. Specifically, we benchmark the training speed on various vision (CIFAR, ImageNet) and language tasks (Shakespeare), with both IID and non-IID partitions, and show DGA can bring $2.55\times$ to $4.07\times$ speedup. Moreover, we built a 16-node Raspberry Pi cluster and show that DGA can consistently speed up real-world federated learning applications.

## 1   Introduction

Federated Learning [18, 27] has gained growing attention as it enables multi-clients distributed training without exposing the data from private users. During the training, only the model updates are exchanged between clients and servers, thus private training data never leave local devices, enhancing privacy. Many successful applications such as next-word prediction [10], voice recognition [38], and health care applications [50] have been derived under the framework.

A significant difference between federated scenarios and typical in-center distributed settings [6, 8] is the networking condition. Unlike high-end in-cluster network infrastructures where high bandwidth (100~Gbps) and low latency ($\leq$1ms) network is available, edge devices are usually connected through wireless and long-distance connection, thus the bandwidth and latency are strictly limited. This dwarfs the performance of federated systems and slows the development of related applications.

While the bandwidth constraints have been efficiently addressed by gradient compression [25], low-rank updates [18] and quantization techniques [43], the issue related to network latency is rarely studied in the recent literature [12, 15]. However, high network latency is inevitable in federated settings because of (1) wireless connections and (2) long-distance transmission (Figure. 1.(b)). On the one hand, the high-density urban office and home environments create a lot of contention, as dozens of devices compete for the same radio frequency. On the other hand, the multi-geographic located data entails a minimum latency cost: even with the speed of light, it requires hundreds of milliseconds to send a packet across the world. In either case, the high latency is a hard barrier introduced by physical limits thus inevitable. If not handled specifically, such communication lag, in the magnitude of hundreds of milliseconds, or even seconds, will significantly degrade the scalability of the learning algorithm as shown in the Figure. 1.

35th Conference on Neural Information Processing Systems (NeurIPS 2021).

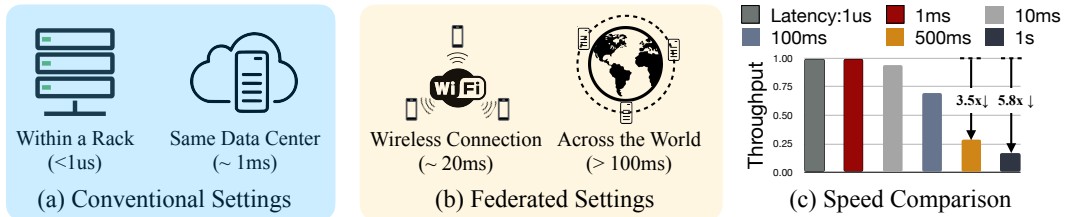

Figure 1: *Left*, *Middle*: The training settings of conventional distributed training v.s. federated learning are very different. *Right*: High latency cost greatly degrades the FedAvg's [27] performance, proposing a severe challenge to scale up the training system. The latency statistics is referenced from Verizon [46] and speedtest [42].

In this paper, we propose **De**layed **G**radient **A**ggregation (DGA) to address the latency bottleneck. The key idea is to delay the gradient averaging to a future iteration, thus the communication can be pipelined with computation. By accepting stale average gradients for model updates, DGA allows the communication to execute in parallel with the computation, thus scalable even under extreme latency. We prove that our DGA shares the same convergence as FedAvg and provide extensive experiments on image and language tasks in both i.i.d. and non-i.i.d settings. We demonstrate that (i) DGA speeds up FedAvg by a factor 2.6× to 4.1× over various datasets; (ii) no accuracy drop under extreme communication lag (e.g., > 1 second). We further set up a Raspberry Pi cluster to simulate the real-world scenario and demonstrate that DGA is robust against network stragglers. Our contributions can be summarized as follows:

- We propose DGA, a novel distributed optimization method to tolerate the communication latency for federated learning. To our best knowledge, our algorithm is the first work that can achieve scalable federated training under high latency (>1s).

- We theoretically justify the convergence of DGA. We show that under reasonable delay interval, it shares the same convergence rate as FedAvg [51]. We also discuss its extension with momentum update to better suit modern federated optimizations.

- We empirically evaluate the accuracy on diverse datasets and benchmark the speed on different latency setups. Under an extremely high latency, DGA can show impressive improvement over previous algorithms while preserving similar performance on both i.i.d and non-i.i.d partitions.

- We build up a Raspberry Pi cluster of 16 devices to evaluate our algorithm in a real-world setting. Even with unpredictable packet loss and latency fluctuation, DGA still shows consistent speedup.

## 2  Delayed Gradient Averaging

### 2.1  Problem Setting, Preliminaries and Related Works

In this work, we formalize the problem as the minimization of sum of stochastic functions,

$$\min_w f(w) = \frac{1}{N}\sum_{i=1}^{N} f_i(w) \quad \text{with} \quad f_i(w) = \mathbb{E}_{\zeta_i}[F_i(w, \zeta_i)],$$

where $N$ denotes the number of clients and $f_i$ represents the loss function on $i$-th client. In the setting of empirical risk minimization, $f_i$ could be further expressed as finite sums and the random variable $\zeta_i$ corresponds to a mini-batch sample.

**Computation and Communication Cost.** Each client represents a computational resource that can either perform local gradient updates or exchange information with others. We model the computation and communication cost by:

- We encounter a computational cost $\tau_g$ for each gradient/mini-batch evaluation.
- The communication suffers a cost $\tau_c$ per transmission. We further define **the normalized communication cost:** $D = \left\lceil \dfrac{\tau_c}{\tau_g} \right\rceil$.

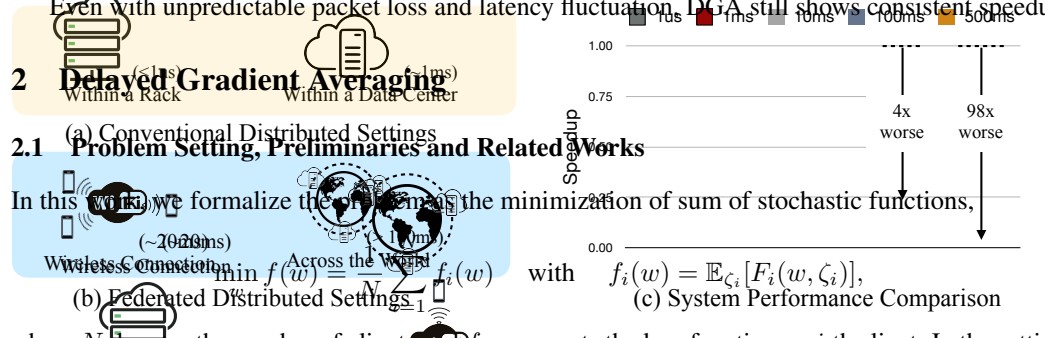

In other words, one communication round takes the same amount of time as $D$ gradient updates. This normalized cost $D$ serves as the main parameter in our discussion on the computation/communication trade-off. Note that the communication cost usually depends on the bandwidth, amount of bits transmitted and the latency, we start revisiting FedAvg [27] to illustrate the effect on latency.

**FedAvg and its variants**   One of the key components in Federated Learning is the step of parameter averaging. This step aggregates the model parameter across clients, without revealing personal training data, enhancing privacy. However, the averaging step is synchronous by design [3], in the sense that no local updates are allowed when the averaging takes place. Specifically, FedAvg [27] alternate the local updates and averaging step:

FedAvg : averaging (communication) $\rightarrow K \times$ local updates $\rightarrow$ averaging (communication) $\cdots$

where the parameter $K$ indicates the number of mini-batches (gradients) locally updated between two averaging steps. The alternating structure of the update naturally decomposes the algorithm into rounds, with $K$ local updates and one communication per round. The total running time of FedAvg after $T$ rounds is given by

$$T_{\text{FedAvg}} = T(K\tau_g + \tau_c) = T\left(K + \frac{\tau_c}{\tau_g}\right)\tau_g \approx T\left(K + D\right)\tau_g. \tag{1}$$

As we can see, the communication cost $D$ inflates the total run time by a non-negligible factor, especially when $D$ is large. For instance, if $D \geq K$, half of the running time is indeed wasted, where clients sit and wait on the synchronization of averages, making the algorithm inefficient. The same drawback persists in all recent extensions of FedAvg [17, 21, 22, 28, 51], its momentum variants [5, 26, 47], variance reduced variant SCAFFOLD [16] and adaptive variant [34], etc.

**Compression/Quantization techniques in FL**   To improve communication efficiency, extensive efforts have been devoted to reducing the bits on gradient exchanges in large-scale distributed training. Techniques such as gradient quantization QSGD [1], Onebit [39], Tengrad [49] and gradient compression DGC [25], Sparsification [48], DenseCommu [43], DoubleSqueeze [44] can safely reduce the amount of information transmitted by a factor of 1000, while maintaining the high performance of the model. These techniques have been successfully applied to accelerate FL [9, 24, 35, 36], enabling training on mobile devices with limited bandwidth connection. Therefore, bandwidth is no longer a critical bottleneck on edge learning.

On the other hand, the latency issue is less discussed in the current FL literature [15] as it is a hard physical barrier. To focus on such inherent and non-improvable subject, **we assume that the bandwidth is sufficient,** and the communication cost $\tau_c$ is dominated by network's latency.** In this case, the running time of FedAvg and its variants in Eq.1 can still be significantly inflated by the latency, even compression/quantization techniques are applied.

The ineffectiveness of the existing approaches to deal with the high latency setting motivates us to ask the following question: is it possible to design an algorithm that could scale up under high latency settings while keeping the accuracy? We provide a positive answer by introducing the delayed gradient averaging.

## 2.2   A step-by-step walk through Delayed Gradient Averaging

The main idea of our algorithm is to allow local updates during the communication of the averaging. In FedAvg, clients send its parameter to each other at the end of each round, wait until the averaging (communication) ended, later starts the next round. In our algorithm, **the averaging barrier is delayed to a later iteration** so that clients can immediately start the next round and **an gradient correction term is designed to compensate the staleness**:

1. Clients send updates to each other at the end of $t$-th round;
2. Clients continue performing local updates using the latest local parameter;
3. When other's $t$-th round info arrived, the client has already performed $D$ extra local updates.
4. Delayed averaging step: replace the local gradients at the $t$-th round by the received averaging (we will detail this step right away).

_______________
*otherwise apply gradient compression techniques.

---

**Algorithm 1** Delayed Gradient Averaging (DGA)

---

1: **Initialize** each worker with $w_{1,1}^i = w_1$ for $i \in [1, N]$, the number of local update $K$, the delayed parameter $D \geq 1$. Define $s = (D-1)//K$ as the integer quotient.
2: **for** rounds $t = 1, \cdots, T$ **do**
3:    **for** client $i$ in parallel **do**
4:       Set $w_{t,1}^i = w_{t-1,K+1}^i$ as the last iterate at round $(t-1)$
5:       **for** $k = 1, \cdots, K$ **do**
6:          Sample the stochastic gradient $g_{t,k}^i$ at the previous iterate $w_{t,k}^i$ and update

$$w_{t,k+1}^i = \begin{cases} w_{t,k}^i - \eta g_{t,k}^i & \text{if } k \not\equiv D \pmod{K} \text{ or } t-1-s < 1; \\ w_{t,k}^i - \eta(g_{t,k}^i - m_{t-1-s}^i + \overline{m}_{t-1-s}) & \text{if } k \equiv D \pmod{K} \text{ and } t-1-s \geq 1. \end{cases}$$

         where $m_{t-1-s}^i$ is the accumulated gradient (see line 8) at the earlier round $t-1-s$, $\overline{m}_{t-1-s}$ is the average of $m_{t-1-s}^i$ among all clients, i.e. $\overline{m}_{t-1-s} = \frac{1}{N}\sum_i m_{t-1-s}^i$.
7:       **end for**
8:       Send the $t$-th round accumulated gradient $m_t^i = \sum_{k=1}^K g_{t,k}^i$ to all other clients.[†]
9:    **end for**
10: **end for**
11: **Return** $\overline{w}_T = \frac{1}{N}\sum_{i=1}^N w_{T,K+1}^i$.

---

The benefit is plain to see: we no longer freeze the local computation power during the communication. To make the discussion more explicit, we denote the parameter on the $i$-th client at the $k$-th iteration within the $t$-th round by $w_{t,k}^i$ and the corresponding stochastic gradient as $g_{t,k}^i$. In the very first round $t = 1$, solely local updates are performed, hence the last iterate in the first round can be expressed as

$$w_{1,K+1}^i = w_{1,K}^i - \eta g_{1,K}^i = \cdots = w_1 - \eta \underbrace{\sum_{k=1}^K g_{1,k}^i}_{:=m_1^i}.$$

As the first round's computation has completed, we send the accumulated gradient $m_1^i := \sum_{k=1}^K g_{1,k}^i$ to all the clients, i.e. perform averaging. Right after the gradients are sent, we immediately proceed the second round's local updates, leaving the first round averages in transmission. By the time the average is received, we already performed $D$ extra local updates in the second round, starting from the last iterate of first round $w_{1,K+1}^i$:

$$\underbrace{w_{2,D}^i - \eta g_{2,D}^i}_{\text{last iterate before averaging}} = \cdots = w_{1,K+1}^i - \eta \underbrace{\sum_{k=1}^D g_{2,k}^i}_{\text{D updates}} = w_1 - \eta \underbrace{\sum_{k=1}^K g_{1,k}^i}_{\text{1st round}} - \eta \underbrace{\sum_{k=1}^D g_{2,k}^i}_{\text{2nd round}}. \tag{2}$$

At this point, the average of the first round arrives

$$\overline{m}_1 = \frac{1}{N}\sum_{i=1}^N m_1^i = \frac{1}{N}\sum_{i=1}^N \sum_{k=1}^K g_{1,k}^i = \sum_{k=1}^K \left(\frac{1}{N}\sum_{i=1}^N g_{1,k}^i\right) := \sum_{k=1}^K \overline{g_{1,k}}, \quad \text{(accumulated averages)}$$

where $\overline{g_{1,k}}$ represents the average gradient at $k$-th iteration in the 1st round. With the averaging $\overline{m}_1$ in hand, we substitute all the first round local gradients in Eq.2 by their averages, leading to:

$$w_{2,D+1}^i = w_1 - \eta \underbrace{\sum_{k=1}^K \overline{g_{1,k}}}_{\text{common on all clients}} - \eta \underbrace{\sum_{k=1}^D g_{2,k}^i}_{\text{2nd round unchanged}} \qquad \text{(delayed gradients averaging)}$$

---

[†]The accumulated gradient is indeed implemented as $m_t \leftarrow m_t + g_{t,k}^i$ under the for loop on $k$, which do not require extra memory to store $g_{t,k}$.

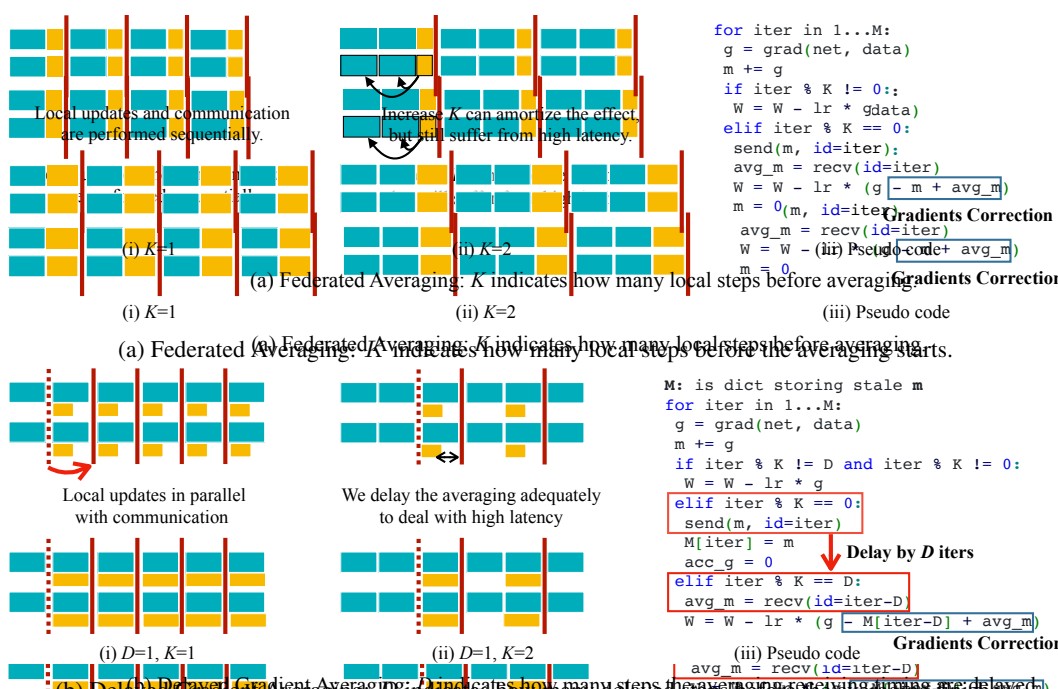

Figure 2: We provide another interpretation of our algorithm DGA, iterating over the gradient iterations, for instance, the $(tK + k)$-th iteration in the pseudo code represents the $t$-th round $k$-th iterations in Algorithm 1. The averaging occurs periodically with period $K$ and the delay parameter $D$ naturally shows up indicating the number of gradients between the sending and reception. The cyan cube in the visualization (a,b) indicates local computation and the yellow cube represents the transmission of the averages. The red bar indicates when the averaging is actually performed. In DGA, the transmission is in parallel to the computation, which is the main reason why DGA can tolerate high latency.

Rewrite the update rule using the accumulated gradients $m_1^i$ and $\overline{m_1}$ gives the compact form:

$$w_{2,D+1}^i = w_{2,D}^i - \eta(g_{2,D}^i \underbrace{-m_1^i + \overline{m_1}}_{\text{gradient correction}}), \qquad \text{(compact DGA update)}$$

serving as the main building block in our Algorithm 1.

In the most general setting, the delayed gradients could take several rounds before arriving. This happens when the latency is high, i.e. $D > K$. To take care of such cases, we express the delay parameter in the quotient/remainder form $D = sK + r$, with $s \geq 0$ and $r \in [1, K]$. The integer quotient $s$ indicates the number of rounds skipped due to the latency. In this scenario, the gradients sent at $t - 1 - s$ round is received in the middle of the $t$-th round, which justifies our update rule in Algorithm 1. The example we discussed earlier in (compact DGA update) is a special case when $s = 0$, i.e. $D \leq K$, where the gradients sent in the last round are received in the current round.

We call such operation as **D**elayed **G**radients **A**veraging (DGA), as every round's averaging has been delayed to later rounds.[‡] With our designed gradient correction term in ( compact DGA update, blue part in Figure. 2b), each update ensures that all clients **replace the previous round's local gradients by their averages**, thus shares the same $(t - D)$-round gradients. In another word, different workers only differ on the most recent $D$ gradients.

Last but not the least, in the ideal case when there is no delay, i.e. $D = 0$, our algorithm DGA recovers the original Federated Averaging [27]. When $D \geq 1$, our algorithm DGA allows the clients to pipeline local updates with communication, and the divergence of Federated Average and DGA is bounded by a constant number. To further clarify, we now conduct a sketch of the convergence analysis by bounding the staleness between different clients.

---

[‡]DGA requires to store the recent $D$ copies of gradients to perform the update. This can bring challenges to edge platforms when the model size is very large.

## 2.3 Theoretical Analysis

To start the convergence analysis, we assume that the objective function is $L$-smooth:

**Assumption 1** ($L$-smoothness). *Each function $f_i(x)$ is L-smooth, i.e. differentiable with L-Lipschitz gradient:*

$$||\nabla f_i(x) - \nabla f_i(y)|| \leq L||x - y||. \quad \forall x, y \in \mathbb{R}^d$$

Remind that each individual function is stochastic and we have access to an unbiased gradient $g^i$ such that $\mathbb{E}[g^i(w)] = \nabla f_i(w)$ for any $w$ and $i$. We further assume that $g^i$ has bounded variance and second moment as in the analysis of FedAvg [51]:

**Assumption 2** (Bounded gradients & variances). *We assume that the unbiased gradients has bounded second moment and variance:*

$$\mathbb{E}||g^i(w)||^2 \leq G^2, \quad and \quad \mathbb{E}||g^i(w) - \nabla f_i(w)||^2 \leq \sigma^2, \quad \forall w, \forall i.$$

We now present two main ingredients to drive the analysis. The first one is that no matter delayed gradient averaging is performed or not, the average parameter across the clients always behaves as a local gradient descent:

**Lemma 2.1.** *Let us denote $\overline{w_{t,k}} = \frac{1}{N} \sum_{i=1}^{N} w_{t,k}^i$, as the average parameters across all clients at t-th round k-th iteration. Then we always have*

$$\overline{w_{t,k+1}} = \overline{w_{t,k}} - \frac{\eta}{N} \sum_{i=1}^{N} g_{t,k}^i.$$

The effect of the delayed averaging step is to align the gradient of the previous round across all clients, without changing $\overline{w_{t,k}}$. In other words, the delayed averaging step reduces the staleness between the clients and ensures the bounded variation given Assumption 2:

**Lemma 2.2** (Bounded Variation). *The difference between the i-th client and the average parameter across all clients is uniformly bounded:*

$$\mathbb{E}\left[||w_{t,k}^i - \overline{w_{t,k}}||^2\right] \leq 4\eta^2(K+D)^2 G^2 \quad \forall t, k, i.$$

Comparing with FedAvg, the upper bound has an additional staleness $D$ due to the delayed gradient averaging. Based on the bound, we can derive the following convergence result:

**Theorem 2.3.** *Under Assumption 1 and 2. The sequence generated by DGA in Algorithm 1 with stepsize $\eta \leq \frac{1}{L}$ satisfies*

$$\frac{1}{TK} \sum_{t=1}^{T} \sum_{k=1}^{K} \mathbb{E}[||\nabla f(\overline{w_{t,k}})||^2] \leq \frac{2}{\eta TK}(\mathbb{E}[f(\overline{w_1})] - f(\overline{w_T})) + 4\eta^2 L^2 G^2(K+D)^2 + \frac{L}{N}\eta\sigma^2$$

Our proof closely follows the analysis of FedAvg in the non-convex setting [51] and we delay it to Appendix A.2. The upper bound on the right-hand-side is essentially the same as the convergence rate of FedAvg, where $(K+D)^2$ taking places of $K^2$ in [51]. Finally, when the number of rounds is large enough, we balance the terms by setting an appropriate stepsize:

**Corollary 2.3.1.** *When the function $f$ is lower bounded with $f(w_1) - f^* \leq \Delta$ and the number rounds $T$ is large enough such that $T \geq N/(K+D)$, then set the stepsize $\eta = \frac{\sqrt{N}}{L\sqrt{T(K+D)}}$ yields*

$$\frac{1}{TK} \sum_{t=1}^{T} \sum_{k=1}^{K} \mathbb{E}[||\nabla f(\overline{w_{t,k}})||^2] = O\left(\frac{2L\Delta + \sigma^2}{\sqrt{NTK}} \cdot \sqrt{1 + \frac{D}{K}} + \frac{N(K+D)}{T}\right).$$

As long as $D = O(K)$ and $N(K+D) \leq T^{1/3}$, the first term dominates and our algorithm shares the same convergence speed $O(1/\sqrt{NTK})$ as the vanilla FedAvg. Meanwhile, as the communication is fully covered by computation, the total run time of DGA is $T_{DGA} = TK\tau_g$, which reduces the run time of FedAvg in (1) by a factor $\frac{K+D}{K}$. In the high latency setting where delay can take multiple rounds, this run time improvement becomes significant.

**DGA with momentum update**   To incorporate momentum update [31, 33] in DGA, we combine the past gradients in an exponential weighted average:

$$u_{t,k}^i = \beta u_{t,k-1}^i + g_{t,k}^i,$$

where $\beta$ is the momentum parameter, usually set as 0.9. Then we use it to perform the local update:

$$w_{t,k+1}^i = \begin{cases} w_{t,k}^i - \eta u_{t,k}^i & \text{if } k \not\equiv D \ (\mathrm{mod}\ K) \text{ or } t-1-s < 1; \\ w_{t,k}^i - \eta(u_{t,k}^i - \frac{1-\beta^D}{1-\beta}(v_{t-1-s}^i - \overline{v_{t-1-s}})]) & \text{if } k \equiv D \ (\mathrm{mod}\ K) \text{ and } t-1-s \geq 1. \end{cases}$$

where $v_t^i = \sum_{k=1}^K u_{t,k}^i$. The delayed gradient averaging step is calibrated carefully by a factor $(1-\beta^D)/(1-\beta)$, due to the exponentially weighted averages. The derivation details are attached in Appendix. We remark that when $\beta = 0$, we recover this to the vanilla update of DGA in Algorithm 1. For other other optimizers (e.g., Adam, RMSProp), such modification can be made similarly. We leave it of as future work.

**Comparison with Asynchronous**   While asynchronous methods remove synchronization lock and deal with heterogeneity between faster and slow workers, it does not take care of the latency issue. We first revisit the traditional ASGD [45]:

$$W : \textbf{pull parameters} \rightarrow \text{local evaluation} \rightarrow \text{push gradients}$$
$$PS : \text{collect gradients} \rightarrow \text{update parameters \& send parameters}$$

Each worker still suffers from high latency when pulling parameters, even though the central parameter server is asynchronous. In contrast, though our delayed averaging has a synchronization barrier (described in Figure 2b), it allows local gradient updates to be performed simultaneously as communication. In other words, the workers (W) keep evaluating new local updates during the process of pulling/pushing parameters. The ability to pipeline communication with computation makes distinct differentiation between ASGD and DGA and allows us to achieve better training efficiency against high latency communication.

**Comparison with FedAvg**   The delayed nature of our averaging step makes it substantially different from the traditional averaging step. In FedAvg, the averaging step synchronizes the weights on all clients, making sure that the next round starts with a common parameter. In contrast, our averaging happens in the middle of the second round, where additional local steps have been proceeded. Due to these $D$ most recent local updates, the parameter $w_{2,D+1}^i$ does not match $w_{2,D+1}^j$ even after averaging, for distinct clients $i \neq j$. The encountered staleness is the trade-off for allowing computation during communication.

**Comparison with SVRG**   For readers who are familiar with variance reduction techniques, the compact DGA update looks very similar to the update in SVRG [13]:

$$w_{t+1} = w_t - \eta\left[\nabla f_i(w_t) - \nabla f_i(\tilde{w}) + \nabla f(\tilde{w})\right], \qquad \text{(SVRG update)}$$

where $\tilde{w}$ is a snapshot reference point and $\nabla f(\tilde{w})$ is the full gradient. The introduction of such reference point in SVRG reduces the variance of stochastic gradient, leading to faster convergence analysis than vanilla SGD.

Although the two formulas share a common pattern, there is an intrinsic difference in the choice of the averaged gradients. In SVRG, the full gradient is evaluated at the reference point $\tilde{w}$, which must be common at all $i$; in contrast, the averaged gradients in our algorithm is not a full gradient by any means, because of the staleness introduced by the extra local updates. Moreover, SVRG uses a fixed reference point $\tilde{w}$ inside the same round while as the vector $\overline{m_t}$ in our algorithm is an accumulation of gradient of the entire round, see (accumulated averages). Hence SVRG and DGA are incomparable from their starting point: while as SVRG aims to reduce variance based on a fixed reference point, DGA is targeting the latency, favoring staleness.

Perhaps the works closest to ours is delayed SGD [2, 14, 23, 52], where the update rule is given by $w_{k+1} = w_k - \eta g_{(k-d)}$. In this case, the gradient updates are delayed but there is no correction

Table 1: Ablation studies about our gradient correction term. Without our correction term, using pure stale gradients suffers from significant accuracy drop. The accuracy is measured on CIFAR-10.

|  | w/o gradient correction | w/ gradient correction |
|---|---|---|
| K=5, D=5 | 88.7 | 89.2 |
| K=5, D=10 | 86.9 | 89.3 |
| K=5, D=15 | 85.5 | 89.0 |
| K=5, D=20 | 84.2 | 88.7 |

term as DGA. Though these studies show sub-linear convergence in theory, such a simple method accumulates the staleness over iterations and hurts model performance especially when the delay steps is large as shown in Table. 1. Therefore, they cannot handle high latency network.

To summarize, we have introduced the delayed gradient averaging, which delays the averaging operation thus allows local updates to be pipelined with communication. We then design a correction term to handle delayed gradients and compensate the staleness. We next conduct experiments showing that DGA speeds up FedAvg without losing accuracy.

# 3 Experiment

## 3.1 Accuracy Evaluation

We evaluate the effectiveness of DGA on diverse tasks: Image classification on CIFAR-10 [19] and ImageNet [20], next word prediction on Shakespheare [41]. We implement DGA in PyTorch framework [30] and choose Horovod [40] as the distributed training backend. The task-specific details are described below.

On CIFAR-10 [19], we train a MobilenetV2-0.25 [37] using 64 workers and each equips with single V100 GPU. The training epochs is 200 and the batch size 64 per worker. The learning rate $\eta$ is initially set to NUM_GPUs $\times$ 0.0125 and momentum $\beta$ is 0.9. The learning rate linearly increases during the first 5 epochs, following the warm-up strategy in [8], and then decays with cosine anneal schedule.

On ImageNet [7], we evaluate ResNet-50 [11] with 64 worker nodes. The total mini-batch size is 2048, and we train the model for 150 epochs. We apply the warm-up strategy in [8] to schedule the learning rate and only random crop and flip are used as data augmentations in the training. The learning rate adopts the same scaling strategy as CIFAR's.

On Shakespeare, we adopt the 2-layer LSTM language model architecture with 1500 hidden units per layer [32] and follow the preprocessing in Leaf [4]. We set the learning rate to 20 and clip gradients with a norm larger than 0.25 to avoid gradient explosion. The model is trained with 40 epochs while the first epoch is used for warm-up.

Table 2: Comparison of FedAvg and our DGA's accuracy on 3 datasets with both i.i.d and non-i.i.d partitions. The speedup is measured on latency with 1s latency. Not only DGA demonstrates consistent training speedup, but also DGA maintains the accuracy, on both i.i.d and non-i.i.d partition.

| Datasets | Partition | FedAvg (K=5) | | FedAvg (K=10) | | FedAvg (K=20) | | DGA (K=5, D=20) | |
|---|---|---|---|---|---|---|---|---|---|
|  |  | Acc | Speedup | Acc | Speedup | Acc | Speedup | Acc | Speedup |
| CIFAR | i.i.d | 88.7 | 1× | 88.5 | 1.51× | 88.1 | 2.05× | 88.6 | 3.16× |
|  | non-i.i.d | 48.2 |  | 47.2 |  | 43.9 |  | 48.0 |  |
| ImageNet | i.i.d | 76.6 | 1× | 76.5 | 1.43× | 76.2 | 1.81× | 76.4 | 2.55× |
|  | non-i.i.d | 55.4 |  | 52.5 |  | 48.6 |  | 54.9 |  |
| Shakespeare | i.i.d | 47.6 | 1× | 47.3 | 1.66× | 47.4 | 2.51× | 47.1 | 4.07× |
|  | non-i.i.d | 36.9 |  | 34.3 |  | 30.1 |  | 36.3 |  |

For non-i.i.d. experiments, we follow the partition used in [4, 22] to split the dataset. In CIFAR and ImageNet, we distribute the dataset such that each device only contains samples from two classes. In Shakespeare dataset, each role is considered as a data source and each device only has two sources.

We show the accuracy and run time comparison when the latency is 1s in Table 2, the speed up is measured in run time, normalized by the run time of FedAvg ($K$=5):

- When we compare FedAvg($K = 5$) with DGA($K = 5, D = 20$), DGA achieves a speed up from $2.5\times$ to $4.07\times$ with almost no accuracy drop.

- To strengthen the baseline, we increase $K$ in FedAvg, which amortizes the effect of latency. However, increasing $K$ reduces the communication frequency, degrading the model performance, especially in the non-i.i.d. setting. As the data distribution is usually non-i.i.d in federated learning, increasing too much $K$ is not a good idea to compensate communication cost.

Therefore, DGA is the clear winner which achieves notable speedup while maintaining accuracy in the high latency setting. This is because DGA **(i) keeps the same communication frequency (ii) pipelines the communication and local updates**, fully covering the latency by local computations. We attach the training curve and ablation studies of different $D$ in Appendix. A.1.

### 3.2 Speedup Comparison

Throughout the derivation of DGA, we have assumed that the latency remains constant. However, in the real-world scenario, latency may vary due to unstable connection and packet losses. One way to handle such potential fluctuation is to set a large delay parameter $D$, pretending the worst scenario happens. In other words, we will artificially set $D$ to be a large number even though the actual latency is small. In this way, we can study how the variation on latency changes the performance of DGA.

In order to simulate different network conditions, we evaluate the effectiveness of DGA in two different ways. We first experiment on a set of synthetic latency controlled by Netem [29], which allows us to control the communication delays precisely. Then we build up a Raspberry Pi cluster consisting of 16 Model 4B+ devices and use Netgear R6300v2 to provide the connection, which reflects a realistic home Wi-Fi environment for federated learning as shown in Figure. 3.

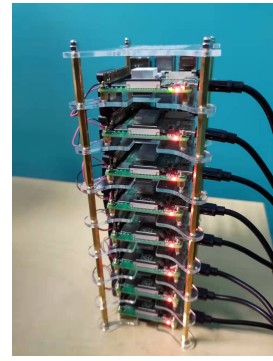

In the Figure. 4a, we manually simulate the network latency from 1ms to 5000ms via Netem [29], and plot the normalized run time. The run time is normalized by FedAvg ($K = 5$) with latency 1ms, which serves as the baseline of in-cluster training. When the latency gradually increases, FedAvg slows down almost linearly. Though setting a larger number of local steps $K$ can alleviate the degradation, the performance gap is still growing. Moreover, let alone a too large $K$ will hurt the final accuracy drastically as shown in Section 3.1. In contrast, DGA (solid red line) as shown in Figure 4a, consistently yields a stable speed as long as the latency is covered by the threshold $D$. Even under an extreme communication delay (e.g., 5 seconds), the training speed is not much affected.

Figure 3: Our Raspberry Pi farm. Experiments are conducted on two racks.

Next we benchmark real-world settings on the Raspberry Pi cluster in the Figure. 4a.. Instead of simulating different latency settings, we consider four representative cases for federated learning [i] wired [ii] (wireless) different rooms [iii] (wireless) different floors [iv] (wireless) different buildings, where the latency numbers are 1ms, 16ms and 132ms and 470ms respectively. By placing the Pi cluster at different locations, we can emulate different quality of connection. Different from synthetic latency, the real wireless connection also suffers from unpredictable network stragglers and packet loss, thus the overall throughput on Pi-clusters is worse and FedAvg's performance degrades more quickly. Even under such challenging network, the throughput of DGA remains steady.

Figure. 5 shows the speedup ratio (scalability) on the Pi cluster. We adopt small models LeNet and 2-layer lstm for the benchmark. Given the limited computation budget of edge devices, the batch size is reduced to 1/8 of the original ones. To better evaluate our algorithm, we choose the most challenging settings where devices are linked through wireless connections from different buildings. With an average latency of 470ms, FedAvg's training is significantly slowed by the communication.

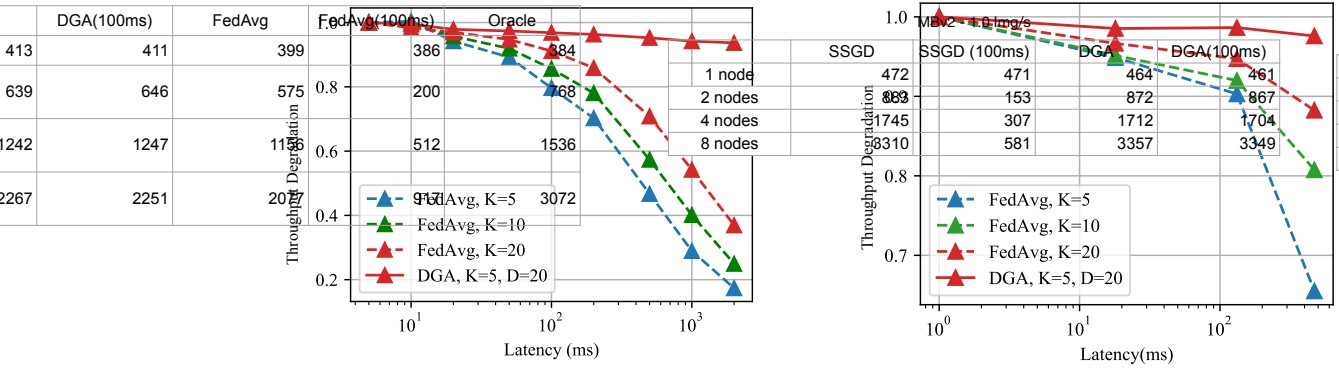

(a) GPU servers with Netem [29] generated synthetic latency from 1ms to 5000ms.

(b) Raspberry Pi cluster (16 nodes) with real-world latency (1ms, 16ms, 132ms, 470ms).

Figure 4: Benchmark FedAvg [27] and DGA on different communication latency. On both latency settings, FedAvg's performance starts to degrade when latency grows, while DGA shows a stable performance even under extreme latency.

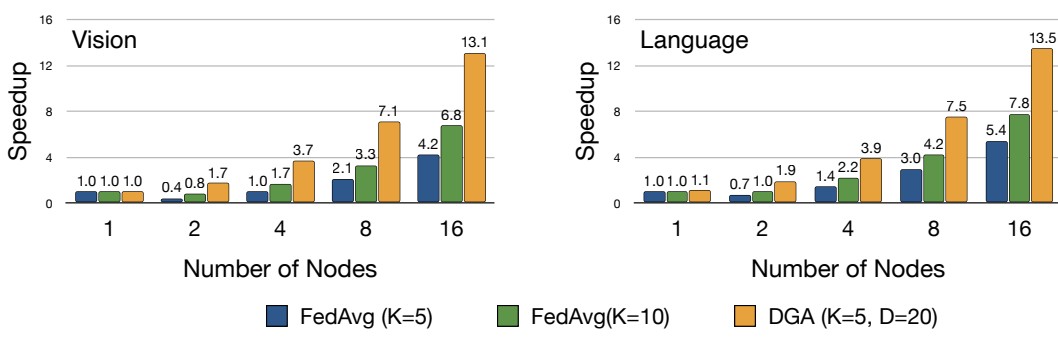

Figure 5: The speedup comparison between FedAvg and DGA on Raspberry Pi cluster. On both vision and language tasks, DGA demonstrate consistent improvement over FedAvg.

When scaling the training to two devices, the speedup ratio is only 0.6, which is even slower than single devices. Instead, our proposed DGA demonstrates ideal scalability under a high-latency network. When scaling to eight devices, the speedup ratio is about 13.1. This performance is close to what conventional algorithms achieved inside a data center.

# 4   Conclusion

In this paper, we propose Delay Gradient Averaging (DGA) to tolerate high network latency in federated learning. We have justified that the theoretical convergence is no slower than FedAvg in non-convex settings. We further demonstrate that our algorithm is capable of enjoying high scalability under poor network conditions and preserving accuracy especially on non-i.i.id partitions. Furthermore, on a realistic setup consisting of 16 Raspberry Pi devices, our algorithms demonstrate consistent speedup previous algorithms. We believe that our work will enable a wide range of federated learning applications in high latency network.

# 5   Acknowledgement

We thank MIT-IBM Watson AI Lab, Samsung, Woodside Energy, and NSF CAREER Award #1943349 for supporting this research. Hongzhou Lin acknowledges that the work is done prior to joining Amazon. Yao Lu acknowledges that the work is done prior to joining Google.

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
