|  | FedAvg | Speedup | DGA(Ours) | Speedup |
|---|---|---|---|---|
| LEAF-CelebA | 81.3 | 1x | 86.8 | 1.6x |
| LEAF-Reddit | 10.4 | 1x | 12.2 | 1.7x |
| LEAF-FEMNIST | 70.3 | 1x | 79.9 | 1.8x |
| LEAF-Shakespeare | 29.3 | 1x | 34.1 | 1.6x |

Table 3: Comparison of FedAvg and DGA on LEAF [4] datasets.

# A  Supplementary material

## A.1  Comparison of Delayed Average and Federated Average

Conventionally, FedAvg may increase the number of local updates to alleviate the effect brought by high latency communication. Though effective as it is, it would lead to obvious performance drop on non-i.i.d partitions because of the reduced communication frequency. On the contrary, DGA only **delays the averaging timing** and **does not decrease the communication frequency** (Figure. 2). Specifically, we sweep both $K$ in FedAvg and $D$ in DGA from 5 to 20. While two algorithms show similar performance on i.i.d scenario (Figure. 6a), DGA's outperforms FedAvg's on non-i.i.d partition (Figure. 6b): The solid lines (DGA) consistently yield better accuracy than the dashed line (FedAvg).

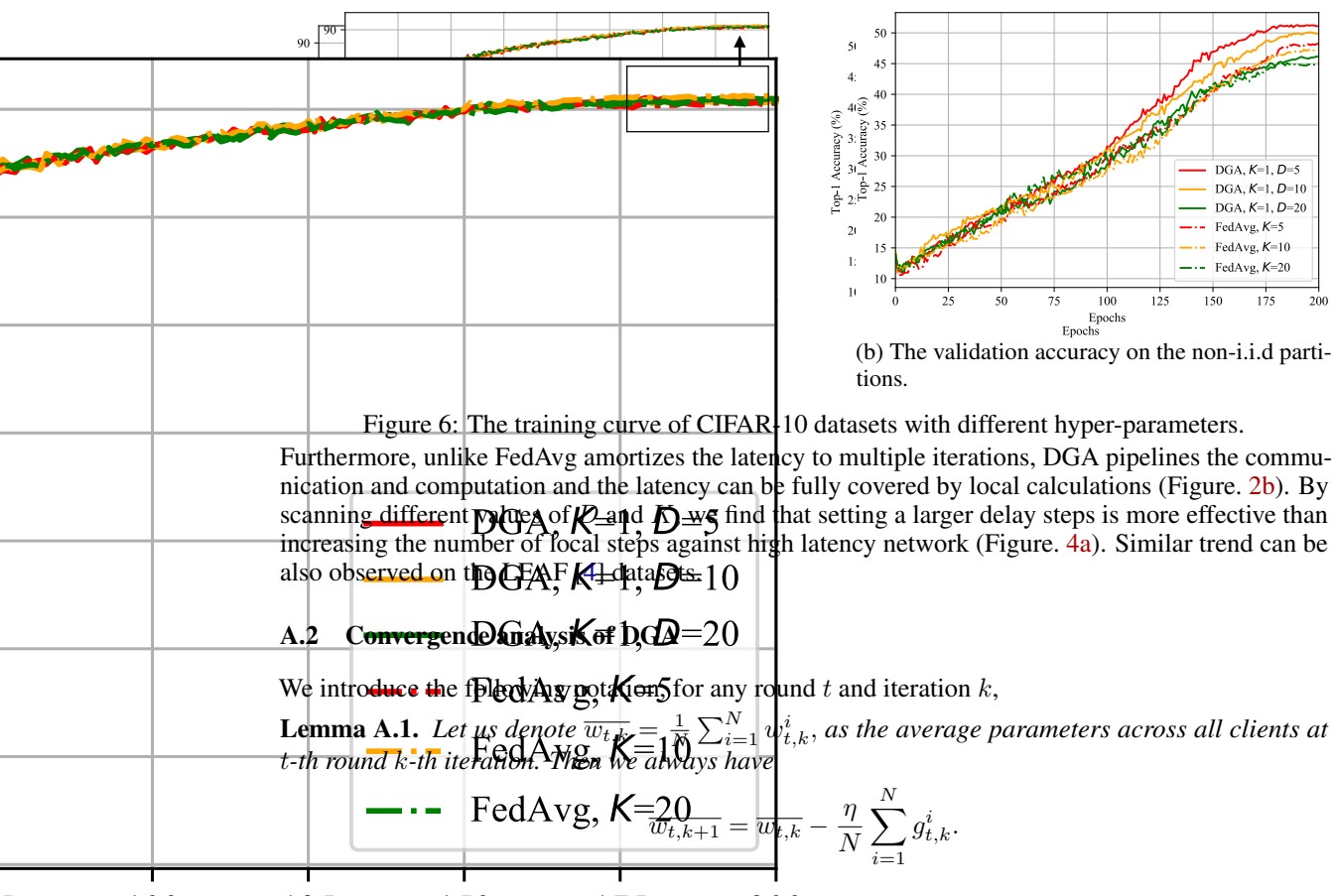

(b) The validation accuracy on the non-i.i.d partitions.

Figure 6: The training curve of CIFAR-10 datasets with different hyper-parameters.

Furthermore, unlike FedAvg amortizes the latency to multiple iterations, DGA pipelines the communication and computation and the latency can be fully covered by local calculations (Figure. 2b). By scanning different values of $K$ and $D$ we find that setting a larger delay steps is more effective than increasing the number of local steps against high latency network (Figure. 4a). Similar trend can be also observed on the LEAF [4] datasets.

## A.2  Convergence Analysis of DGA

We introduce the following notations for any round $t$ and iteration $k$,

**Lemma A.1.** *Let us denote* $\overline{w_{t,k}} = \frac{1}{N}\sum_{i=1}^{N} w_{t,k}^i$, *as the average parameters across all clients at $t$-th round $k$-th iteration. Then we always have*

$$\overline{w_{t,k+1}} = \overline{w_{t,k}} - \frac{\eta}{N}\sum_{i=1}^{N} g_{t,k}^i.$$

*Proof.* This is clearly true when $k \neq D \pmod{K}$. When the delayed averaging is performed, the statement holds by noticing as $\overline{m_{t-1-s}} = \frac{1}{N}\sum_{i=1}^{N} m_{t-1-s}^i$ □

**Lemma A.2** (Bounded Variation). *The difference between the $i$-th client and the average parameter is uniformly bounded:*

$$\mathbb{E}\left[\|w_{t,k}^i - \overline{w_{t,k}}\|^2\right] \leq 4\eta^2(K+D)^2 G^2 \quad \forall t, k, i.$$

*Proof.* If $k \geq r$, then the gradients up to $t - 1 - s$ round are aligned

$$w_{t,k}^i = \overline{w_{t-1-s}} - \underbrace{\eta m_{t-s}^i}_{t-s \text{ round}} - \cdots - \underbrace{\eta m_{t-1}^i}_{t-1 \text{ round}} - \underbrace{\eta \sum_{j=1}^{k} g_{t,j}^i}_{t \text{ round}}$$

As each round consists of $K$ local updates, the number of local gradients not aligned is $sK + k = sK + r + (k - r) = D + (k - r) \leq D + K$. A similar reasoning provides the same bound when $k < r$. This means the parameter $w_{t,k}^i$ and $w_{t,k}^j$ differs by at maximum $2(K + D)$ local gradients throughout the entire process. Hence

$$\mathbb{E}\left[\|w_{t,k}^i - w_{t,k}^j\|^2\right] \leq 4\eta^2 (K + D)^2 G^2,$$

and finally the desired bound follows by Jensen's inequality. $\qquad\square$

**Theorem A.3.** *Under Assumption 1 and 2. The sequence generated by DGA in Algorithm 1 with stepsize $\eta \leq \frac{1}{L}$ satisfies*

$$\frac{1}{TK} \sum_{t=1}^{T} \sum_{k=1}^{K} \mathbb{E}[\|\nabla f(\overline{w_{t,k}})\|^2] \leq \frac{2}{\eta TK}(\mathbb{E}[f(w_1)] - f^*) + 4\eta^2 L^2 G^2 (K + D)^2 + \frac{L}{N}\eta\sigma^2$$

*Proof of Theorem 2.3.* By the $L$-smoothness of $f$, we have

$$\mathbb{E}[f(\overline{w_{t,k+1}})] \leq \mathbb{E}[f(\overline{w_{t,k}}) + \langle \nabla f(\overline{w_{t,k+1}}), \overline{w_{t,k+1}} - \overline{w_{t,k}} \rangle + \frac{L}{2}\|\overline{w_{t,k+1}} - \overline{w_{t,k}}\|^2].$$

From Lemma A.1, we have

$$\mathbb{E}[\|\overline{w_{t,k+1}} - \overline{w_{t,k}}\|^2] = \eta^2 \mathbb{E}[\|\frac{1}{N}\sum_{i=1}^{N} g_{t,k}^i\|^2]$$

$$= \eta^2 \mathbb{E}[\|\frac{1}{N}\sum_{i=1}^{N}\left(g_{t,k}^i - \nabla f_i(w_{t,k}^i)\right)\|^2] + \eta^2 \mathbb{E}[\|\frac{1}{N}\sum_{i=1}^{N}\nabla f_i(w_{t,k}^i)\|^2]$$

$$= \frac{\eta^2}{N^2}\sum_{i=1}^{N}\mathbb{E}[\|g_{t,k}^i - \nabla f_i(w_{t,k}^i)\|^2] + \eta^2 \mathbb{E}[\|\frac{1}{N}\sum_{i=1}^{N}\nabla f_i(w_{t,k}^i)\|^2]$$

$$\leq \frac{\eta^2\sigma^2}{N} + \eta^2 \mathbb{E}[\|\frac{1}{N}\sum_{i=1}^{N}\nabla f_i(w_{t,k}^i)\|^2].$$

Moreover,

$$\mathbb{E}[\langle \nabla f(\overline{w_{t,k}}), \overline{w_{t,k+1}} - \overline{w_{t,k}} \rangle]$$

$$= -\eta \mathbb{E}[\langle \nabla f(\overline{w_{t,k}}), \frac{1}{N}\sum_{i=1}^{N} g_{t,k}^i \rangle]$$

$$= -\eta \mathbb{E}[\langle \nabla f(\overline{w_{t,k}}), \frac{1}{N}\sum_{i=1}^{N}\nabla f_i(w_{t,k}^i) \rangle]$$

$$= \frac{\eta}{2}\mathbb{E}[\|\nabla f(\overline{w_{t,k}}) - \frac{1}{N}\sum_{i=1}^{N}\nabla f_i(w_{t,k}^i)\|^2] - \frac{\eta}{2}\mathbb{E}[\|\nabla f(\overline{w_{t,k}})\|^2] - \frac{\eta}{2}\mathbb{E}[\|\frac{1}{N}\sum_{i=1}^{N}\nabla f_i(w_{t,k}^i)\|^2]$$

The first term can be upper bounded using Lemma A.2 with the $L$-smoothness condition:

$$\mathbb{E}[\|\nabla f(\overline{w_{t,k}}) - \frac{1}{N}\sum_{i=1}^{N}\nabla f_i(w_{t,k}^i)\|^2] = \mathbb{E}[\|\frac{1}{N}\sum_{i=1}^{N}(\nabla f_i(\overline{w_{t,k}}) - \nabla f_i(w_{t,k}^i)\|^2]$$

$$(\text{Jensen's inequality}) \leq \frac{1}{N}\sum_{i=1}^{N}\mathbb{E}[\|(\nabla f_i(\overline{w_{t,k}}) - \nabla f_i(w_{t,k}^i)\|^2]$$

$$(L\text{-smoothness}) \leq \frac{L^2}{N}\sum_{i=1}^{N}\|\overline{w_{t,k}} - w_{t,k}^i\|^2$$

$$(\text{Lemma A.2}) \leq 4\eta^2 L^2 G^2 (K+D)^2.$$

Now, combining the above inequalities together with $\eta \leq \frac{1}{L}$ yields

$$\mathbb{E}[f(\overline{w_{t,k+1}})] - \mathbb{E}[f(\overline{w_{t,k}})]$$

$$\leq \frac{L\eta^2 - \eta}{2}\mathbb{E}[\|\frac{1}{N}\sum_{i=1}^{N}\nabla f_i(w_{t,k}^i)\|^2] - \frac{\eta}{2}\mathbb{E}[\|\nabla f(\overline{w_{t,k}})\|^2] + 2\eta^3 L^2 G^2 (K+D)^2 + \frac{L}{2N}\eta^2\sigma^2$$

$$\leq -\frac{\eta}{2}\mathbb{E}[\|\nabla f(\overline{w_{t,k}})\|^2] + 2\eta^3 L^2 G^2 (K+D)^2 + \frac{L}{2N}\eta^2\sigma^2.$$

Rearrange the terms yields

$$\mathbb{E}[\|\nabla f(\overline{w_{t,k}})\|^2] \leq \frac{2}{\eta}\left(\mathbb{E}[f(\overline{w_{t,k}})] - \mathbb{E}[f(\overline{w_{t,k+1}})]\right) + 4\eta^2 L^2 G^2 (K+D)^2 + \frac{L}{N}\eta\sigma^2$$

Telescoping from $t = 1, ..., T$ and $k = 1, .., K$ yields

$$\frac{1}{TK}\sum_{t=1}^{T}\sum_{k=1}^{K}\mathbb{E}[\|\nabla f(\overline{w_{t,k}})\|^2] \leq \frac{2}{\eta TK}\left(\mathbb{E}[f(w_1)] - f^*\right) + 4\eta^2 L^2 G^2 (K+D)^2 + \frac{L}{N}\eta\sigma^2$$

$$\square$$

**Corollary A.3.1.** *When the function $f$ is lower bounded with $f(w_1) - f^* \leq \Delta$ and the number rounds $T$ is large enough such that $T \geq N/(K+D)$, then set the stepsize $\eta = \frac{\sqrt{N}}{L\sqrt{T(K+D)}}$ yields*

$$\frac{1}{TK}\sum_{t=1}^{T}\sum_{k=1}^{K}\mathbb{E}[\|\nabla f(\overline{w_{t,k}})\|^2] = O\left(\frac{2L\Delta + \sigma^2}{\sqrt{NTK}} \cdot \sqrt{1 + \frac{D}{K}} + \frac{N(K+D)}{T}\right).$$