# OpenReview forum: "Delayed Gradient Averaging: Tolerate the Communication Latency for Federated Learning"
_NeurIPS.cc/2021/Conference — NeurIPS 2021 Poster_

### Official Review · Reviewer_e8gp · 2021-07-16

**Rating:** 7
**Confidence:** 4

**Summary:**

This paper propose a delayed gradient averaging method to hide communication latency behind computation in federated learning scenario to achieve better speedup. Theoretical and experimental results all show its advantage over FedAvg in some scenarios. The authors also carefully design simulate real federated learning environment to evaluate their method.

**Limitations And Societal Impact:**

The authors should conduct more experiments with different number of workers, sync periods, and normalized communication cost, to show the true shortcomings and advantages of their method.

**Main Review:**

The overall writing of this paper is good and the description of their method is pretty clear.  The idea itself is simple but the author gives a detail convergence analysis. And the authors also design their experiments carefully. I have several questions and suggestions:
1. It is well known that with the increasing of parallel workers, the accuracies of models trained by model-averaging like method will degrade a lot. In this paper, only models trained with 8 workers are evaluated. According to my experience, 8-worker is a pretty safe distributed training setting. Could you also give the model accuracies of other configs (1, 2, 4,16,32 ...) to compare the accuracy degradation levels of FedAvg and DGA? With these experiments, we can gain more insight about the true strength and shortcomings of DGA over FedAvg.
2. In the main paper, the K for DGA is set to 5 and in the supplementary material, it is set to 1. But for FedAvg, K=5 10 and 20 are tried. Does it mean that DGA only works for small sync period due to the staleness introduced by delayed averaging ?  Acturally more frequent communication will bring additional money cost in real scenarios. Could you also add some experiments to show the influence of "D" ?
3.  For distributed training with momentum, an important pioneer work is missing:
Chen K, Huo Q. Scalable training of deep learning machines by incremental block training with intra-block parallel optimization and blockwise model-update filtering[C]//2016 ieee international conference on acoustics, speech and signal processing (icassp). IEEE, 2016: 5880-5884.
 SlowMo extends this work to more tasks.


**Time Spent Reviewing:**

2

---

> ### Author Response · Authors · 2021-08-10
> **Response to Reviewer e8gp**
>
> We thank reviewer e8gp's useful comments and would like to respond as follows:
>
> ## Compare DGA and FedAvg accuracies of larger number of workers?
> In Table 1, we evaluate the accuracy of DGA and FedAvg on both IID and non-IID partitions with **64** workers and we further attach a detailed accuracy comparison with more devices. Below we show the results from CIFAR-non-iid and MobilenetV2-0.25: DGA consistently demonstrates better performance than FedAvg when two algorithms yield similar training speed.
>
> |                    | FedAvg | Speedup | DGA (ours)     | Speedup |
> | ------------------ | ------ | ------- | -------------- | ------- |
> | #Num of Workers=8  | 46.1%  | 1.0x    | 50.1% (+4.0%)  | 3.41x   |
> | #Num of Workers=16 | 43.9%  | 1.0x    | 47.6% (+3.7%)  | 3.26x   |
> | #Num of Workers=32 | 38.1%  | 1.0x    | 43.2% (+5.1%)  | 3.10x   |
> | #Num of Workers=64 | 33.2%  | 1.0x    | 40.5% (+7.3% ) | 2.98x   |
>
> In addition, we have added experiments with **128 workers** on the LEAF dataset[4] below
>
> |                  | FedAvg | Speedup | DGA (ours)  | Speedup |
> | ---------------- | ------ | ------- | ---- | ------- |
> | LEAF-CelebA      | 81.3   | 1x      | 86.8 | 2.3x    |
> | LEAF-Reddit      | 10.4   | 1x      | 12.2 | 1.7x    |
> | LEAF-FEMNIST     | 70.3   | 1x      | 79.9 | 1.8x    |
> | LEAF-Shakespeare | 29.3   | 1x      | 34.1 | 1.6x    |
>
>
> The 8 raspberry pi devices are used for a real-world speed benchmark. We follow the number of clients as in previous studies (FedProx[1]: 10 clients per round) to set up our raspberry pi cluster. While we would like to benchmark on more devices, setting up hundreds/thousands of raspberry pis is beyond the ability of most research groups. Thus we use GPU with synthetic latency (details described in Section 3.2) to simulate the environment. Table 1 and the above newly added LEAF experiments have shown the speedup with **64 and 128 workers** respectively.
>
>
> ## Does it mean that DGA only works for a small sync period due to the staleness introduced by delayed averaging?
>
> This is indeed a misunderstanding. While previous methods have to set a larger $K$ to reduce communication overhead (at the sacrifice of model precision), our DGA can tolerate the high network latency without increasing K (thus performing better on real-world non-iid data). In the main paper, we set K=5 to be consistent with FedAvg’s baseline.
>
> The proposed delayed averaging ($D$) and periodical averaging ($K$) are orthogonal techniques thus in the supplementary we intended to test on different values to compare their performance. We are sorry that it creates confusion for the reviewer. In below, we provide a more detailed comparison on both $D$ and $K$ to show that DGA works for various sync periods.
>
>
> ##  Experiments to show the influence of "D" ?
>
> A more detailed comparison of $D$ and $K$ is shown below, where the percentage is the evaluated accuracy (higher the better), time is the real-measured communication overhead (lower the better). We adopt the CIFAR dataset and mobilenetv2-0.25 architecture as same in Table 1 row 1.
>
> |   | FedAvg    | DGA(D=5)     | DGA(D=10)    | DGA(D=20)    |
> | ---- | ------------- | ------------ | ------------ | ------------ |
> | K=5  | 44.1% (477ms) | 44.4% (85ms) | 43.7% (31ms) | 43.2% (30ms) |
> | K=10 | 41.9% (265ms) | 41.5% (62ms) | 40.9% (24ms) | 40.4% (25ms) |
> | K=20 | 33.1% (161ms) | 33.1% (43ms) | 33.0% (20ms) | 32.7% (21ms)|
>
>
> An insightful observation is that scaling $D$ (DGA) is much more effective than $K$ (FedAvg) when a federated system aims for high training throughput. In terms of training overhead, we notice that scaling $D$ will quickly reduce the overhead from 447ms to 31ms while the accuracy only drops by less on percent. Adjusting $D$ (conventional approach), though also reduces the overhead, sacrifices the accuracy by a neat 10% on the non-iid dataset.
>
> ## Missing Reference
> Thanks for sharing the pointer. SlowMo[2] is already included in our related work and we will add the reference of Chen. K 2016[3] in the revision.
>
> We wish that our response has addressed your concerns, and turns your assessment to the positive side. If you have any questions, please feel free to let us know during the rebuttal window. We appreciate your suggestions and comments!
>
> Reference:
> * [1]: Tian Li, Anit Kumar Sahu, Manzil Zaheer, Maziar Sanjabi, Ameet Talwalkar, Virginia Smith. Federated Optimization in Heterogeneous Networks. In MLSys 2020.
> * [2]: Jianyu Wang, Vinayak Tantia, Nicolas Ballas, Michael Rabbat. SlowMo: Improving Communication-Efficient Distributed SGD with Slow Momentum. In ICLR 2020.
> * [3]: Kai Chen,Qiang Huo. Scalable training of deep learning machines by incremental block training with intra-block parallel optimization and blockwise model-update filtering. In ICASSP 2016.
> * [4] Sebastian Caldas, Sai Meher Karthik Duddu, Peter Wu, Tian Li, Jakub Konečný, H. Brendan McMahan, Virginia Smith, and Ameet Talwalkar. LEAF: A Benchmark for Federated Settings.

---

> > ### Comment · Reviewer_e8gp · 2021-08-25
> > **Thanks for your response**
> >
> > The authors have addressed all my concerns with more detailed experimental evaluation. I really appreciate their effort and useful invention. I would like to raise my score to 7.

---

> > > ### Author Response · Authors · 2021-08-27
> > > **Response to Reviewer e8gp**
> > >
> > > Dear Reviewer e8gp,
> > >
> > > Many thanks for reviewer e8gp's updates, and the acknowledgment of our efforts in the rebuttal. We are glad that our response has solved your concern. We will update the paper with your suggested experiments with different number of workers, influence with $D$, as well as the missing reference of Chen. K 2016[1]. Again we are very thankful for your effort and support!
> > >
> > >
> > > [1] Kai Chen,Qiang Huo. Scalable training of deep learning machines by incremental block training with intra-block parallel optimization and blockwise model-update filtering. In ICASSP 2016.
> > >
> > >
> > > Bests,
> > >
> > > Authors

---

### Official Review · Reviewer_XyoU · 2021-07-16

**Rating:** 6
**Confidence:** 4

**Summary:**

The work proposes to overlap the local gradient computation step on each client in a federated ML setup with the latency cost associated with communicating the averaged gradients from all clients. This is done by allowing the clients to proceed with computation of local gradients on K mini-batches for a new round (t+1) using the latest local parameter until the gradient average from the previous round (t) is received with a delay (aka, *Delayed Gradient Averaging*). In doing so, the client is said to have performed D extra local updates which are combined with the delayed gradient average. The work is said to be the first of its kind to consider high latency (>1s) situations to perform scalable federated learning without sacrificing performance.

**Limitations And Societal Impact:**

*Authors must clarify and highlight the limitations of the work in the manuscript as claimed in the checklist.*

**Main Review:**

**Originality:**
The work seems to belong to the field of asynchronous communication within Federated Machine Learning setups with a rich body of literature. However, the paper does not mention this direction as part of related work and how the proposed DGA framework is novel and unique. The introduction of communication cost D is interesting to characterize the number of additional local updates a client makes in each round.

**Quality:**

The proposed work provides theoretical guarantees for convergence and how it is possible to recover the vanilla FedAVG [28] and SVRG [13]. The experiments support the theoretical analysis. Experiments with Raspberry Pi are a good contribution to characterize real-world latency effects on various algorithms. The authors mention in the checklist that they discuss the limitations of their work. It will be better if they can highlight it in the manuscript as I could not find the discussion.

**Clarity:**

1. Please clarify how this work is different from asynchronous communication setups to bring the novelty to work
2. What is the relationship between periodic synchronization (averaging, denoted as K? as per Fig 2 caption) and delayed averaging period in this work? Is there a way to optimally compute the delayed averaging period and what should it be dependent on?

- Line 264:  should be Figure 3b
- Lines 235-236: must mention this is for Shakespeare dataset as per Table 1
- Lines 272-280: Discussion is provided for Fig 4 (mobile net) while the reader is expected to draw the same analysis for LSTM. The paper must make similar references to the other figure


**Significance:**
The work is interesting as it tries to show a simple delayed averaging could help overlap computation and communication in federated setups. I am not sure how it is different from asynchronous setups and the impact of staleness on final performance. A literature survey is missing for asynchronous FL setups.

----------------------------
**After Authors's Comments**

The above comparison between asynchronous and DGA must be a part of your manuscript to clarify the difference and misunderstanding the readers might face. The timing diagram further helps to understand better. Please use **Pipelining** as this is a proper technical term for overlapping and also gives more clarity to folks from Systems research to put your work in context.

*Authors must clarify and highlight the limitations of the work in the manuscript as claimed in the checklist.*

I am happy to increase my score from 4 to 6.


**Time Spent Reviewing:**

2

---

> ### Author Response · Authors · 2021-08-10
> **Response to Reviewer XyoU**
>
> We appreciate reviewer XyoU's feedback, the comments and questions are very relevant. On the other hand, we also realize there is some misunderstanding on our method. Please allow us to clarify them in more detail.
>
> ## How this work is different from asynchronous communication setups?
>
> We address the reviewer's concern from two aspects (a) asynchronous method **CANNOT** handle the latency issue; (b) DGA is not asynchronous, but can **fully cover the communication by computation** by allowing local updates during transmission.
>
> First, we would like to clarify that existing asynchronous algorithms cannot address the latency issue. Let us revisit the traditional ASGD [1, 2] is a sequential process between parameter server (PS) and local workers (W):
> * (W): **pull parameters** → local updates → push gradients.
> * (PS): once received gradients → update model parameters.
>
> Each worker still suffers from high latency when pulling parameters, even though the central parameter server is asynchronous. While asynchronous methods deal with heterogeneity between faster and slow workers, it **does not** take care of the latency issue.
>
> In contrast, our method delayed gradient averaging (DGA) is synchronous, in the sense, there is always a synchronization barrier (described in Figure 2.b). Beyond the synchronous aspect, another major difference between DGA and ASGD is that we perform local gradient updates simultaneously as communication. In other words, the workers (W) **keep evaluating new local updates** during the process of pulling/pushing parameters. Intuitively, this means the parameter on the primary server is always $D$ iterations behind the parameter on the local machine. (assuming communication cost = $D$ local iterations)
>
> The reviewer does make a good catch that both DGA and ASGD **incorporate stale gradients**, which is their similarity. The major advantage of DGA lies in the ability to **pipeline communication with computation**, which never blocks the local machine. This property allows us to achieve better computation/communication efficiency where communication is fully overlapped with computation. This naturally leads to the reviewer's second question regarding the delayed averaging update.
>
>
> ## What is the difference between periodic synchronization and delayed averaging?
>
> To illustrate the difference, it is necessary to introduce the axis of time.  Let us start with a periodic synchronization with $K = 3$, i.e. synchronization every $3$ iterations:
>
> ```
>   	          -- send and recv parm. --
> 	            ↑	                    ↓
> T_1 => T_2 => T_3                       T_4 => T_5 => T_6
> ```
> During the synchronization, **all the local machines are blocked** to wait for the synchronization to finish. No local update is allowed.
>
>
> For delayed averaging, let's further assume that $D = 3$, meaning the time of communication is equivalent to $3$ iterations of local updates. **The local worker keep performing local updates while the (averaging) parameters are in transmission**:
>
> ```
>   	      send parm. ---------- recv parm.
> 	           ↑                     ↓
> T_1 => T_2 => T_3  => T_4 => T_5 => T_6
> ```
> In particular, by the time averaging parameter is received, we are already at iter=6 locally. The delayed averaging keeps T4--T6 unchanged while replacing T1-T3 by their averages (described in line 6 of Algorithm 1).
>
> As a consequence, even after delayed averaging, the weight on $i$-th client is still different than the one on $j$-th client, due to the fact that T4--T6 step on $i$-th client is different than the T4--T6 step on $j$-th client.
>
>
> ## How the proposed DGA framework is novel and unique?
>
> The strength of the DGA framework is to **fully overlap communication with computation**. We believe this is what an ultimately distributed optimization algorithm looks like.
>
> To achieve this goal, it is necessary to tolerate **stale gradients** and enable **local updates**. To the best of our knowledge, we are not aware of any existing approach that allows both of them simultaneously. The major challenge lies in **allowing local updates along with stale gradients**.
>
> We believe our method is a solid step towards efficient distributed algorithms under high latency, where communication is fully overlapped with computation. We believe this is a major conceptual step barely taken previously.
>
> ## Is there a way to optimally compute the delayed averaging period and what should it be dependent on?
> The optimal D is determined by computation time per iteration and network latency:
> $D_{\text{optimal}} = \lceil\frac{\text{network latency}}{\text{computation time per local update}}\rceil$
> In this case, the communication cost is fully covered by the computation cost. If the network suffers from fluctuation / packet loss, an upper bound estimation on the network latency is required.
>
> ## Writing Suggestions:
> We have revised the typo, writing and description accordingly. We will update them in the final version.
>
> We wish that our response has addressed your concerns, and turns your assessment to the positive side. If you have any questions, please feel free to let us know during the rebuttal window. We appreciate your suggestions and comments!

---

> > ### Comment · Reviewer_XyoU · 2021-08-26
> > **Thank you for Clarifications**
> >
> > Dear Authors,
> >
> > The above comparison between asynchronous and DGA must be a part of your manuscript to clarify the difference and misunderstanding the readers might face. The timing diagram further helps to understand better. Please use **Pipelining** as this is a proper technical term for overlapping and also gives more clarity to folks from Systems research to put your work in context.
> >
> > *Authors must clarify and highlight the limitations of the work in the manuscript as claimed in the checklist.*
> >
> > I am happy to increase my score from 4 to 6.

---

> > > ### Author Response · Authors · 2021-08-27
> > > **Response to Reviewer XyoU**
> > >
> > > Dear Reviewer XyoU,
> > >
> > > Many thanks for your updates and acknowledge of our efforts the in the rebuttal. We are glad that our response has solved the confusion about asynchronous and DGA. We will use the proper techinical term **Pipelining** in next version to make better clarity. We apologize that the limitation is not highlighted in the paper and discuss as follows:
> > >
> > > [**Limitation 1: Extra Storage**]: To perform delayed update, DGA requires to store the recent $D$ copies of gradients (line 6 in algorithm 1, and line 87 in our [attached code](https://anonymous.4open.science/r/NeurIPS-2633-Delayed-Gradient-Averaging/tools/delayed_sync.py)). This may brings chanllenges to edge platforms when model size is very large.
> > >
> > > [**Limitation 2: Engineer Efforts to Integrate DGA with New Optimizers**]: Though DGA proposes a general idea to tolerate high network latency, it is not easy to integrate with new optimizers. For example, to support momentum (line 267 in paper) we rework the formula and the code. It takes some effort to support all popular optimizers like Adam, AdaMax, RMSProp and etcs.
> > >
> > > We promise that the stated changes will be delivered in the final version. Again we are veryful thankful for your time and effort to help strengthen our work!
> > >
> > > Bests,
> > >
> > > Authors

---

### Official Review · Reviewer_9p8v · 2021-07-19

**Rating:** 5
**Confidence:** 3

**Summary:**

This paper presents a new distributed federate learning method, DGA, to overcome long communication latency. The authors provide theoretical analysis about DGA's convergence rate, which shows that DGA makes same convergence as FedAvg, and empirically show that DGA can tolerate long network latency without hurting the accuracy. DGA delays the gradient averaging to a future iteration so that the communication can be overlapped with local update computation, which enables scalability even under long latency and network stragglers. In detail, in DAG, a training is divided into communication rounds, and each round consists of K local updates and one synchronization (gradient averaging). DGA allows gradient averaging at i-th round to D-th updates in a future round. DGA achieves high speed-up without hurting the accuracy by allowing the same communication frequency while reducing latency by controlling D knob.

**Ethical Concerns:**

This paper does not have any ethical concerns.

**Limitations And Societal Impact:**

Yes.

**Main Review:**

Thank you for submitting your work to NeurIPS 2021. Here is a summary of strengths of the paper.

[+] This paper tries to address an important problem (long network latency bottleneck) in federate learning (enable efficient learning under long network latency)

[+] algorithm description is easy to understand

[+] experiment with Raspberry Pi for real world wireless communication environment.

But, I have a few serious concerns about the paper.

[-] It looks that the technical depth is not mature enough for publication . The idea of overlapping communication for synchronization at i-th round and computation at (i+1)th round and allowing stale gradient values looks quite straightforward.

[-] It seems that authors' assumption about network BW and latency might hold for very limited domains/use cases. How many application domains.use cases the author's assumptions about BW and latency, BW is enough but latency is long, hold for?

[-] Experiments with 8 workers looks insufficient. It would be great if the authors could provide experiment results with larger number of workers than 8.

**Time Spent Reviewing:**

4.5

---

> ### Author Response · Authors · 2021-08-10
> **Reponse to Reviewer 9p8v**
>
> We thank reviewer 9p8v for the informative feedback. We would like to point out several original aspects of our method which might not shine out in the first place. Please allow us to clarify them in more detail.
>
>
> ## DGA's Technical Contribution
> We would like to clarify the contribution of DGA as follows:
>
> * We appreciate the reviewer considering the idea of overlapping communication and computation to be natural. We believe this is what an ultimately distributed optimization algorithm looks like. To achieve this goal, it is necessary to enable both **1) stale gradients** and **2) local updates**. To the best of our knowledge, we are not aware of any existing approach that allows both of them simultaneously:
>
>     + In the federated learning literature[1], a lot of efforts have been put to innovate local updates, usually followed by a weight averaging (synchronization), hence there are no stale gradients.
>     + In the distributed optimization literature, the stale gradients are usually incorporated under the **asynchronous** setting. While tremendous variants have been proposed regarding the heterogenous update on the side of the parameter server, the local machines are blocked during the transmission. Hence there are no local updates.
>
>     The major challenge lies in allowing **local updates along with stale gradients**.
>
> * The key to overcome this challenge is the step of delayed gradients averaging. This makes DGA **fundamentally different from the traditional FedAvg.** Specifically, right after the traditional FedAvg, all the local machine shares the same weights. If we stick to such a concept, it is impossible to pipeline averaging with local updates. In contrast, in our delayed gradients averaging, the averaging executes in parallel with local model evaluations. Here is a graphical example:
>     ```
>               send param. ------------ recv param.
>                    ↑                     ↓
>     T_1 => T_2 => T_3  => T_4 => T_5 => T_6
>     ```
>      The local worker keeps performing local updates while the averaging parameters are in transmission, thus DGA can tolerate a high latency without slowing the training.
>      Though the weight on $i$-th client is different than the one on $j$-th client at each iteration, we prove that the divergence is bounded and will not degrade the model performance both in theory and practice (Section 2.3).
>
> Based on the above reasons, we believe our method is a solid first step towards latency-tolerant distributed algorithms, where communication is fully covered by computation. This is a major conceptual step barely taken previously.
>
>
> ## Assumption about network bandwidth and latency
>
> We completely agree with the reviewer that in a realistic setting, both bandwidth and latency become poor and critical for edge devices. **The bandwidth issue is well-solved with recent techniques, but latency remains unsolved,** thus we propose DGA.
>
> A lot of gradient compression/quantization has been proposed to address the bandwidth issue: One-bit [2] and Ternary grad [3] suggest a reduction of bits from 32 (float) to 1 or 2 is possible, which leads to 16~32x times bandwidth saving. Moreover, gradient compression [4] techniques further show that we can compress the transferred weights up to 700x without sacrificing accuracy. With these techniques in hand, which we have carefully mentioned in our related work section, bandwidth is no longer critical even in some extreme cases.
>
> Concretely, for our real-world raspberry pi setup, the model contains 1.4M parameters and takes 1/3s forward and backward, a minimum bandwidth of 125 Mbps suffices to support the training. This is not an excessive requirement to be satisfied: 2.4G wifi ($\sim$150 Mbps), 5G wifi ($\sim$1300 Mbps), cellular network 4G LTE ($\sim$150Mbps) and 5G ($\sim$10Gbps), let alone we can still apply gradient quantization / compressing technique to further reduce the transferred bits.
>
> In contrast, the latency issue is rarely discussed and remains unsolved. There are very few techniques available to reduce the influence of latency in the current literature, especially when the latency is in the scale of seconds. In our experiments, we observe that the latency is near 500ms and a byte may take up to 2 seconds to deliver when there are packet losses and network fluctuation. This is about physical limits and is hard to improve. Our main contribution is on the communication delay, thus we have focused on the influence of latency. We will add discussions about the assumption to make it more clear.
>
>
> ## Experiments with the number of workers larger than 8.
>
> We would like to emphasize that **our synthetic experiments in Table 1 and Figure 3.1 are evaluated under 16 & 64 workers**. The 8 worker case is only for the real-world Raspberry-Pi experiment. Unfortunately we have no access to a large cluster of Raspberry-Pi. Manually setting up a larger Raspberry-Pi network goes beyond our hardware management capacity. We also want to point out that we have already taken a step further than most of the FL paper, by including such an experiment. Though 8 Raspberry-Pis are a small-scale benchmark, it is a minimum evaluation under the real-world environment, not just the synthetic ones. We are very willing to experiment with more Pi devices if affordable resources are available.
>
>
> In addition, we have added experiments with **128 workers** on the LEAF dataset[5] below
>
> |                  | FedAvg | Speedup | DGA (ours)  | Speedup |
> | ---------------- | ------ | ------- | ---- | ------- |
> | LEAF-CelebA      | 81.3   | 1x      | 86.8 | 2.3x    |
> | LEAF-Reddit      | 10.4   | 1x      | 12.2 | 1.7x    |
> | LEAF-FEMNIST     | 70.3   | 1x      | 79.9 | 1.8x    |
> | LEAF-Shakespeare | 29.3   | 1x      | 34.1 | 1.6x    |
>
> We further conduct ablation studies using different numbers of workers to evaluate the accuracy, benchmarked on CIFAR and MobilenetV2-0.25:
>
> |                    | FedAvg | Speedup | DGA (ours)     | Speedup |
> | ------------------ | ------ | ------- | -------------- | ------- |
> | #Num of Workers=8  | 46.1%  | 1.0x    | 50.1% (+4.0%)  | 3.41x   |
> | #Num of Workers=16 | 43.9%  | 1.0x    | 47.6% (+3.7%)  | 3.26x   |
> | #Num of Workers=32 | 38.1%  | 1.0x    | 43.2% (+5.1%)  | 3.10x   |
> | #Num of Workers=64 | 33.2%  | 1.0x    | 40.5% (+7.3% ) | 2.98x   |
>
>
> And based on synthetic results on 64 and 128 workers, we certainly believe that the DGA can scale up to more Raspberry-Pis when applicable.
>
>
> We wish that our response has addressed your concerns, and turns your assessment to the positive side. If you have any questions, please feel free to let us know during the rebuttal window. We appreciate your suggestions and comments!
>
> References:
> * [1] Peter Kairouz, H Brendan McMahan et al. Advances and open problems in federated learning.
> * [2]: Frank Seide, Hao Fu, Jasha Droppo, Gang Li, Dong Yu. 1-Bit Stochastic Gradient Descent and Application to Data-Parallel Distributed Training of Speech DNNs. In InterSpeech 2014.
> * [3]: Wei Wen, Cong Xu, Feng Yan, Chunpeng Wu, Yandan Wang, Yiran Chen, Hai Li. TernGrad: Ternary Gradients to Reduce Communication in Distributed Deep Learning. In NIPS 2017.
> * [4]: Yujun Lin, Song Han, Huizi Mao, Yu Wang, William J. Dally. Deep Gradient Compression: Reducing the Communication Bandwidth for Distributed Training. In ICLR 2018.
> * [5] Sebastian Caldas, Sai Meher Karthik Duddu, Peter Wu, Tian Li, Jakub Konečný, H. Brendan McMahan, Virginia Smith, and Ameet Talwalkar. LEAF: A Benchmark for Federated Settings.

---

> > ### Author Response · Authors · 2021-09-01
> > **Response to Reviewer 9p8v**
> >
> > Dear Reviewer 9p8v,
> >
> > We appreciate your time for the review, and we hope to have a follow-up conversation with reviewer 9p8v to see if our response addresses your concerns.
> >
> > We would appreciate it if reviewer 9p8v could comment to the most important points in our rebuttal. For example, **pipelining stale gradients and local updates** to make DGA **fundamentally different** from the traditional FedAvg and async algorithms. Reviewer XyoU thought the timing diagram **clearly illustrates the novelty of DGA** and thus **increase the score from 4 to 6**. We would like to hear your feedback to check whether this solves your concerns as well.
> >
> > We sincerely hope that reviewer 9p8v could kindly check our response. Thank you very much!
> >
> > Bests,
> >
> > Authors

---

### Official Review · Reviewer_ZCJE · 2021-07-25

**Rating:** 7
**Confidence:** 4

**Summary:**

Authors propose a method that allows communication in federated learning to run in parallel with local gradient computation; practically removing the communication delay. Authors provide theoretical convergence bounds and compare their method with two different but similar algorithms (only in writing). Authors provide formulation for both with and without momentum on local gradient updates.
 Authors also provide experimental analysis on non-FL datasets (CIFAR, ImageNet, Shakespeare) that are sharded to adapt to FL settings. Authors also provide results from an implementation of the algorithm on eight raspberry pi devices.

**Limitations And Societal Impact:**

this work has no societal impact or limitations.

**Main Review:**

Although paper suggests federated learning as the target problem, authors fail to provide an algorithm (or an experimentation result of one) that can be used in a real world fl setting. Here are some of the concerns I have:
* In FL, local datasets can (and most likely will) have different number of examples, assuming that all clients go through K local updates is far from reality (this setting is only applicable to distributed training)
* In my opinion, running FL algorithms on non-FL datasets is no longer acceptable for FL papers, I'd suggest running experiments with FL datasets like the ones in LEAF.
* Authors have not considered weighted averaging between clients in FL.
* The proposed method is not compatible with approaches that provide alternative losses for local devices, e.g. FedProx.
* In the algorithm 1, authors mention that clients send gradients to each other in the communication phase, this is more close to peer-to-peer learning, in FL a central server maintains the model and handles communications.
* Authors assume that all clients participate in all rounds, in real world settings usually there are many clients and a small sub set of them are selected for each round.
* Authors do not mention privacy which should be at least discussed in every FL paper.
* Assuming that all clients have the same computation power  is also an impractical assumption, i.e. local gradient computation takes Tg time for all clients
* Although I was pleasantly surprised to see that the authors have tried their algorithms on Raspberry Pi devices, only doing the experiment on 8 devices is again more similar to what happens in distributed training rather than in FL where there are usually at least thousands of devices.

I would suggest that authors re-write the paper having distributed training (or cross-silo FL) as the target problem and propose their solution for when the delay between the compute nodes is considerable, alternatively authors can address the comments and adapt their algorithm to FL settings.

**Time Spent Reviewing:**

2

---

> ### Author Response · Authors · 2021-08-10
> **Response to Reviewer ZCJE**
>
> We thank Reviewer ZCJE for the thoughtful comments and suggestions. We address each of your questions below.
>
> ## FL Datasets:
>
> For the language task, we are using the **exactly same** dataset as LEAF-Shakespeare (the one provided by the LEAF project) [1] and the same preprocessing for the Next-Character Prediction. For CIFAR and ImageNet, though the original dataset is not designed for FL scenarios, we follow the same partition strategy as LEAF-Synthetic. This strategy is also adopted in previous studies [2, 3, 4]. Experiments on non-iid partitions are also included (in Figure 1).
>
> As suggested by the reviewer, we now include new experiments on other LEAF datasets with **128 clients**:
>
> |                  | FedAvg | Speedup | DGA(Ours) | Speedup |
> | ---------------- | ------ | ------- | --------- | ------- |
> | LEAF-CelebA      | 81.3%  | 1.0x    | 86.8%     | 2.3x    |
> | LEAF-Reddit      | 10.4%  | 1.0x    | 12.2%     | 1.7x    |
> | LEAF-FEMNIST     | 70.3%  | 1.0x    | 79.9%     | 1.8x    |
> | LEAF-Shakespeare | 29.3%  | 1.0x    | 34.1%     | 1.6x    |
>
> Column 1 and 3 indicates the test accuracy and Column 2 and 4 indicates the relative speedup. The configurations of FedAvg and DGA are the same as in Table 1. We see that DGA consistently outperforms FedAvg in terms of both end-to-end training time as well as model accuracy.
>
>
>
> ## Weighted Averaging
>
> Weighted averaging is important since local datasets may have different numbers of examples. DGA is a general algorithm and can be easily extended to support this. Concretely, in algorithm1, we can modify DGA to the following to support weighted averaging:  Firstly, instead of only sending the gradients, we also send out the weighted factor $b_{i, j}$
>
> $ send(g_{t,k}^{i}) => send(g_{t,k}^{i}, b_{t,k}^{i}) $
>
> where $i$ denotes the worker index,  $j$ denotes the outer iteration and $k$ denotes current local steps. Then we can update the weights as
>
> $ w_{t,k+1}^{i} = w_{t,k}^{i} - \mu (g_{t,k}^{i} +  M_{t - 1 - s}^{'i} - \overline{M'_{t - 1 - s}}) $
>
> where $B_{t}^{i}= \sum_{j} b_{t,k}^{i}$, $M_{t}^{'i} = \frac{1}{B_{t}^{i}} \sum_{k=1}^{K} b_{t,k}^{i} g_{t,k}^{i}$ and $\overline{M_{t}^{'i}} =  \sum_{i} M'^{i}_{t}$
>
>
> ## FedProx
> DGA is a general algorithm that focuses on how to pass/handle stale gradients, thus compatible with most loss functions. For the mentioned FedProx, the loss term defined in the original FedProx paper [5] to find best suitable $w_{t,k+1}^{i}$ is
>
> $  h(w_{t,k+1}^{i}; w_{t,0}^{i})  = F (w_{t,k+1}^{i}) + \frac{\lambda}{2} ||w_{t,k+1}^{i} - w_{t,0}^{i}||^2 $
>
> A straightforward way to enable the proximal term is to set the prox center as the $D$-th iterates instead of the $0$-th iterate. Though a more careful analysis is needed to guarantee the theoretical convergence, we do think such an extension is plausible.
>
> While there are a lot of improvements and follow-up variations of FedAvg like FedProx[5], FedNova[4],FedGP[6] and many more,  we are unable to cover all of them in one paper. Here we choose the FedAvg as our baseline since it is the first and most widely used federated optimization.
>
> ## Central-PS v.s. Peer-to-peer
> In the paper, we use peer-to-peer communication to make the presentation simple. DGA can be also applied in central server infrastructure, just as FedAvg. In short, the algorithm goes as
>
> ```
> Parameter Server:
>   for i:= 0 to iter_max:
>     if i % K == 0:
>       [run in background / pipelined]
>       collect i^th round gradients and average
>       send the averaged results back
>
> Worker:
>   for i:= 0 to iter_max:
>     local evaluation to compute gradients
>     if i % K == D:
>         receive (i - d)^th averaged gradients
>     if i % K == 0:
>       send i^th local gradients to the parameter server
>     Update the models with local gradients and stale averaged ones (if applicable).
> ```
>
> ## Assume all clients have the same computation power
> Our paper focuses on the latency issue, thus we assume all devices to be the same to ease discussion. For unbalanced computational power, one quick adaptation is to set different batch sizes: Powerful workers with larger batch sizes and weak workers with small batch sizes, so that the $T_{\text{local computation}}$ across different workers would be similar. The modified algorithm is similar to the aforementioned weighted averaging case.
>
> ## Limited number of PI devices
>
> We would like to clarify that our evaluation consists of two parts, accuracy and speed. For the accuracy comparison, we have already evaluated **64 workers** in the paper and we further conduct ablation studies with different numbers of works as shown in below:
>
> |                    | FedAvg | Speedup | DGA (ours)     | Speedup |
> | ------------------ | ------ | ------- | -------------- | ------- |
> | #Num of Workers=8  | 46.1%  | 1.0x    | 50.1% (+4.0%)  | 3.41x   |
> | #Num of Workers=16 | 43.9%  | 1.0x    | 47.6% (+3.7%)  | 3.26x   |
> | #Num of Workers=32 | 38.1%  | 1.0x    | 43.2% (+5.1%)  | 3.10x   |
> | #Num of Workers=64 | 33.2%  | 1.0x    | 40.5% (+7.3% ) | 2.98x   |
>
>
> For the speed benchmark, we follow the number of clients as in previous studies (FedProx[5]: 10 clients per round) to set up our raspberry pi cluster. While we would like to benchmark on more devices, setting up hundreds/thousands of raspberry pis is beyond the ability of most research groups. Thus we use GPU with synthetic latency (described in Section 3.2) to simulate the environment. Table 1 in the paper (64 workers) and the below newly added LEAF experiments (128 workers) have shown our speedup. DGA consistently outperforms FedAvg in terms of end-to-end training time, while demonstrating better accuracy on non-iid datasets. We believe this is enough to exhibit the advantage of DGA.
>
> |     | FedAvg | Speedup | DGA(Ours) | Speedup |
> | ---------------- | ------ | ------- | ---- | ------- |
> | LEAF-CelebA      | 81.3   | 1x      | 86.8 | 1.6x    |
> | LEAF-Reddit      | 10.4   | 1x      | 12.2 | 1.7x    |
> | LEAF-FEMNIST     | 70.3   | 1x      | 79.9 | 1.8x    |
> | LEAF-Shakespeare | 29.3   | 1x      | 34.1 | 1.6x    |
>
> ## Select a subset at each training phase
> DGA can support selecting a subset at each training phase as long as there are enough devices to run the training process.  We will update the algorithm description.
>
> ## Privacy Discussion
> DGA will enable more on-device training under a high-latency network, thus users no longer need to upload data to the cloud and privacy can be protected. We will add related discussion into the main paper.
>
>
> References:
> * [1] Sebastian Caldas, Sai Meher Karthik Duddu, Peter Wu, Tian Li, Jakub Konečný, H. Brendan McMahan, Virginia Smith, and Ameet Talwalkar. LEAF: A Benchmark for Federated Settings.
> * [2] Xiang Li, Kaixuan Huang, Wenhao Yang, Shusen Wang, Zhihua Zhang. On the Convergence of FedAvg on Non-IID Data. In ICLR 2020.
> * [3] Xiaoxiao Li, Meirui Jiang, Xiaofei Zhang, Michael Kamp, Qi Dou. FedBN: Federated Learning on Non-IID Features via Local Batch Normalization. In ICLR 2021.
> * [4] Jianyu Wang, Qinghua Liu, Hao Liang, Gauri Joshi, H. Vincent Poor. Tackling the Objective Inconsistency Problem in Heterogeneous Federated Optimization. In NeurIPS 2020.
> * [5] Tian Li, Anit Kumar Sahu, Manzil Zaheer, Maziar Sanjabi, Ameet Talwalkar, Virginia Smith. Federated Optimization in Heterogeneous Networks. In MLSys 2020.
> * [6] Minxue Tang, Xuefei Ning, Yitu Wang, Yu Wang, Yiran Chen. FedGP: Correlation-Based Active Client Selection for Heterogeneous Federated Learning.

---

> > ### Comment · Reviewer_ZCJE · 2021-08-25
> > **Good answers (+ good experiments) but not detailed**
> >
> > Adding LEAF experiments has improved the quality of the paper, good job!
> > concerns still remaining:
> > * If FL is the targeted application, all algorithms need to reflect that. re "we use peer-to-peer communication to make the presentation simple"
> > * 128 clients per round is a good starting number, experimentation needs to expand to a 1000 clients.
> > * re "thus users no longer need to upload data to the cloud and privacy can be protected.", privacy is not protected in this way at all. White/Black box attacks on the client uploaded model can still uncover/approximate sensitive client data.
> >
> > I'd like to keep the score for now, unless I get clear answers on the above questions.

---

> > > ### Author Response · Authors · 2021-08-27
> > > **Response to Reviewer ZCJE**
> > >
> > > # Response to Reviewer ZCJE
> > >
> > > Dear Reviewer ZCJE,
> > >
> > > Thanks for the acknowledgement of our new results in the rebuttal! We would like to address your new question as follows:
> > >
> > >
> > > ## Algorithm needs to reflect that FL is targeted application
> > >
> > > We agree that central-client communication is more suitable for FL scenarios and we will update all algorithm:
> > >
> > > For Figure 2.a.iii, we will change it to
> > > ```
> > > for T:= 1 to iter_max:
> > >   randomly select a subset of total devices
> > >   for client i in parallel
> > >     for K:= 1 to iter_max:
> > >       local evaluation to compute gradients
> > >       accmulate the local gradients
> > >       update model with the local gradients
> > >     send accmulated local gradients to the parameter server
> > >     reset the accmulation buffer
> > >     receive averaged results and update model
> > > ```
> > >
> > > For Figure 2.b.iii, we will change it to
> > >
> > > ```
> > > for T:= 1 to iter_max:
> > >   randomly select a subset of total devices
> > >   for client i in parallel
> > >     for K:= 1 to iter_max:
> > >       local evaluation to compute gradients
> > >       accmulate the local gradients
> > >       if k % K == D:
> > >           receive averaged results and update model
> > >       else:
> > >           update model with the local gradients
> > >     send accmulated local gradients to the parameter server
> > >     reset the accmulation buffer
> > > ```
> > >
> > > These algorithm changes will be delivered in our final version.
> > >
> > > ## Expanding to more than 1000 clients
> > >
> > > Our work focuses on real-world scenarios and our implementation **requires one device / accelerator for each client** as shown in the [attached code](https://anonymous.4open.science/r/NeurIPS-2633-Delayed-Gradient-Averaging). Unlike previous studies where a device / accelerator simulates multiple clients (e.g., [FedProx](https://github.com/litian96/FedProx/blob/master/flearn/trainers/fedprox.py#L36)) to only verify  the accuracy, expanding the number of devices to larger than 1000 in our experiments will require a thousand GPUs / Raspberry Pis. This is beyond our hardware capacity.
> > >
> > > Though we are unable to report the training throughput on 1000 devices, we are now working on re-writing the code to support simulated run to compare the accuracy. We will try out best to deliver the results during the rebuttal window。
> > >
> > >
> > > ## Privacy Protection
> > >
> > > We agree that white/black box attacks may reveal the sensitive data from the uploaded models. It is a known problem for most federated learning algorithms [1, 2, 3], not just for DGA. There have been techniques to alleviate the problem (e.g., noisy gradient, sparse/quantized gradients, differential privacy, homomorphic encryption). These methods are orthogonal to DGA and can be applied together to improve the security.
> > >
> > > Please let us know if you have any more comments / questions. Again we are very thankful for your effort and support!
> > >
> > >
> > > * [1] Ligeng Zhu, Zhijian Liu, Song Han. Deep Leakage from Gradients. In NeurIPS 2019.
> > > * [2] Jonas Geiping, Hartmut Bauermeister, Hannah Dröge, Michael Moeller. Inverting Gradients--How easy is it to break privacy in federated learning. In NeurIPS 2020.
> > > * [3] Hongxu Yin, Arun Mallya, Arash Vahdat, Jose M. Alvarez, Jan Kautz, Pavlo Molchanov. See through Gradients: Image Batch Recovery via GradInversion. In CVPR 2021.

---

> > > > ### Comment · Reviewer_ZCJE · 2021-08-27
> > > > **Some of responses are not clear**
> > > >
> > > > * Paper as I see it revised is a good example fo cross-silo FL (not a cross-device FL, as in this latter case as I mentioned there are many many devices (tens-to-hundreds of millions), a few (could be a few thousands) devices are [randomly] selected for each round to be trained, this means there is a small chance that a client is selected many times.) Please mention this.
> > > >
> > > > * Implications of differential privacy are not completely orthogonal, e.g. think about gradient clipping in DP. I think it is better to admit the lack of work in an area and mention that clearly in the paper as opposed to making claims like:  "These methods are orthogonal to DGA" without detailed studies.
> > > >
> > > > I think adding more experiments will definitely help the quality of the paper and I thank the authors for committing to that.

---

> > > > > ### Author Response · Authors · 2021-09-02
> > > > > **Response to Reviewer ZCJE**
> > > > >
> > > > > ## Simulated Experiments with thousand devices
> > > > >
> > > > > As promised earlier, we have added new results containg 1024 devices. We simulate 64 clients on a single GPU and totally use 16 cards to obtain results of 1024 devices. The results of non iidand iid CIFAR-10 (same as Table 1 row 2 in the main paper) are listed below to compare the accuracy.
> > > > >
> > > > > * On non-IID partitions
> > > > >     | Number of Clients=1024  | FedAvg        | DGA(D=5)     | DGA(D=10)    |
> > > > >     | ------------------- | ------------- | ------------ | ------------ |
> > > > >     | K=5                 | 26.2%  | 26.0%  | 26.7%  |
> > > > >     | K=10                | 23.4% | 23.1%  | 23.2%  |
> > > > >
> > > > > * On IID partitions
> > > > >     | Number of Clients=1024  | FedAvg        | DGA(D=5)     | DGA(D=10)    |
> > > > >     | ------------------- | ------------- | ------------ | ------------ |
> > > > >     | K=5                 | 88.3%  | 88.1%  | 88.0%  |
> > > > >     | K=10                | 87.9% | 88.0%  | 87.7%  |
> > > > >
> > > > > An insightful observation is scaling $D$ (DGA) is better than  $K$ (FedAvg) to preserve the accuracy, especially on non-iid partitions and it also yields higher training throughput as we shown previous studies.
> > > > >
> > > > > ## Implications on differential privacy
> > > > >
> > > > > We thank the reviewer for the careful inspection, and we admit that our current writing focuses much on the efficiency and have not taken the privacy discussion into account. We will follow your suggestion to improve this and additional effort is required to enhance the privacy protection.
> > > > >
> > > > > ## Cross-silo FL
> > > > > Thanks for your suggestion. We agree that cross-silo FL is a better example for DGA usage. We are reworking on the related sections and will clarify this in the final version.
> > > > >
> > > > > Once again, we thank you for your construtive suggestions and positive comments which help strengthen our work.
> > > > >
> > > > > Bests,
> > > > >
> > > > > Authors

---

> > > > > > ### Comment · Reviewer_ZCJE · 2021-09-02
> > > > > > **increasing my score by 1**
> > > > > >
> > > > > > Thanks to the authors, I am now convinced that this paper is acceptable.

---

### Author Response · Authors · 2021-08-10
**General Response**

# General Response
We sincerely appreciate all reviewers’ time and efforts in reviewing our paper. We truly thank for the constructive suggestions which will strengthen our paper. Yet, we would like to stretch reviewers' attention to our contribution and a common (misunderstood) criticism in the experimental setting.



## Main Contributions
The main novelty of our method is to **fully overlap communication and computation**, achieving better efficiency under a high latency network. We believe this is a major conceptual step barely taken previously. We decide to present the paper in an FL setting because it is where long communication latency is inevitable (signal channel contention / long physical distance) and undermined (up to hundreds of milliseconds or even seconds). We believe our method is a solid first step towards latency-tolerant distributed algorithms.

## A common concern (misunderstood)
One common criticism regarding our experimental setting is that only 8 workers are used in the evaluation. We believe there is some misunderstanding.

Indeed, our experiment consists of two parts: **synthetic environment and real-world Raspberry-Pi environment**. While our Raspberry-Pi experiments consist of 8 devices,  for the synthetic experiments, **CIFAR/LEAF-Shakespeare are evaluated on 16 workers while ImageNet is evaluated using 64 workers** (as described in line 218 and 222 on page 7 of the paper). We are very willing to benchmark experiments with thousands of Raspberry-Pi, unfortunately we just do not have access to a such larger-scale Raspberry-Pi cluster. **We emphasize that 8 Raspberry-Pis are a minimum evaluation under the real-world environment, and this is already a step further than most of the FL papers.**

To incorporate the reviewer's suggestion, we have performed additional experiments including:

- Evaluation on a larger number of devices (128 workers in synthetic setting) [ZCJE, 9p8v, e8gp];
- Experiments on LEAF datasets [ZCJE];
- Ablation study on different $D$ and $K$ combinations [e8gp]

We hope our responses below could clarify all reviewers’ confusion and alleviate all concerns.

---

### Author Response · Authors · 2021-08-22
**Does the rebuttal address your concerns?**

Dear Reviewers,

We sincerely appreciate your time for the review, and we really hope to have a further discussion.

Could you kindly check our responses and let us know whether our rebuttal addresses your concerns?

Thanks a lot,

Authors

---

### Author Response · Authors · 2021-09-03
**Summary of our rebuttal**

Dear all reviewers and ACs,

We sincerely thank you for the time, efforts and valuable suggestions to further improve our work during the rebuttal.  We genuiely appreciate the positive **7-7-6-5** scores from **ZCJE**, **e8gp**, **XyoU** and **9p8v**.

Specifically,
* We thank reviewer **ZCJE** for appreciating the novelty of our real-world experiements on Raspberry pis, and detailed suggestions on experiments and writing. We appreciate **ZCJE** for increasing the score **from 6 to 7**.
* We thank reviewer **e8gp** for compliments on the clarity of the paper and the novelty of the idea. We are thankful for **e8gp** increasing the score **from 5 to 7** after reading our rebuttals.
* We thank reviewer **XyoU** for recoginizing the novelty and simpleness of our algorithm, as well as the helpful writing suggestions. We appreciate **XyoU** for the postive re-consideration **from 4 to 6** after rebuttal.
* We regret for not hearing the feedback from **9p8v** during the rebuttal window and we sincerely hope to have a further discussion to see whether our response addressed the raised questions. We believe that our rebuttal should clarify the concerns and we would like to answer additional questions if there are more.


The new added experiments, promised discussion about privacy and limitation, and improved manuscripts will be delivered in our final version. Once again, thank all reviewers’ and ACs' time and efforts!

Bests,

Authors

---

### Decision · Program_Chairs · 2021-09-27

**Decision:**

Accept (Poster)

**Comment:**

This paper proposes an approach to make Federated training more efficient by pipelining communication and computation. In particular, clients continue performing local updates on a previous model while they are communicating with the server. In addition to some theoretical convergence analysis for the smooth non-convex setting, the paper provides empirical evidence supporting the promise of this approach both in simulation and in an implementation using Raspberry PI’s. The approach seems especially well-suited for cross-silo federated learning when clients have unreliable communication channels or communication latency is otherwise a bottleneck.

While the reviewers raised some significant concerns in their initial reviews, the author responses largely addressed these in a satisfactory way, leading multiple reviewers to raise their scores. As a result, I’m happy to recommend that we accept this paper. When preparing the camera ready version, the authors are expected to implement the suggestions and corrections that were discussed during the rebuttal period. In particular, it is important to:
* Include the additional experiments (LEAF, CIFAR with 1k devices) either in supplementary material, or by moving other material to supplementary material, as you see fit,
* Clarify terminology and concepts (e.g., using “pipelining”, differentiating from asynchronous methods)
* To include clarifications about the implementation using a centralized parameter server vs. peer-to-peer synchronization
* Clarify the limitations around privacy guarantees, additional storage, and challenges around supporting other optimizers
* Adding additional references suggested be reviewers